# Dynamics of embryonic stem cell differentiation inferred from single-cell transcriptomics show a series of transitions through discrete cell states

**Sumin Jang[1,2]\*[†], Sandeep Choubey[1,2][†], Leon Furchtgott[1,3], Ling-Nan Zou[1], Adele Doyle[1,2], Vilas Menon[4], Ethan B Loew[1,2], Anne-Rachel Krostag[4], Refugio A Martinez[4], Linda Madisen[4], Boaz P Levi[4], Sharad Ramanathan[1,2,4,5,6]\***

[1]FAS Center for Systems Biology, Harvard University, Cambridge, United States; [2]Department of Molecular and Cellular Biology, Harvard University, Cambridge, United States; [3]Biophysics Program, Harvard University, Cambridge, United States; [4]Allen Institute for Brain Science, Seattle, United States; [5]School of Engineering and Applied Sciences, Harvard University, Cambridge, United States; [6]Harvard Stem Cell Institute, Harvard University, Cambridge, United States

**\*For correspondence:**
suminjang@gmail.com (SJ);
sharad@cgr.harvard.edu (SR)

[†]These authors contributed equally to this work

**Competing interests:** The authors declare that no competing interests exist.

**Abstract** The complexity of gene regulatory networks that lead multipotent cells to acquire different cell fates makes a quantitative understanding of differentiation challenging. Using a statistical framework to analyze single-cell transcriptomics data, we infer the gene expression dynamics of early mouse embryonic stem (mES) cell differentiation, uncovering discrete transitions across nine cell states. We validate the predicted transitions across discrete states using flow cytometry. Moreover, using live-cell microscopy, we show that individual cells undergo abrupt transitions from a naïve to primed pluripotent state. Using the inferred discrete cell states to build a probabilistic model for the underlying gene regulatory network, we further predict and experimentally verify that these states have unique response to perturbations, thus defining them functionally. Our study provides a framework to infer the dynamics of differentiation from single cell transcriptomics data and to build predictive models of the gene regulatory networks that drive the sequence of cell fate decisions during development.

## Introduction

During differentiation, cells repeatedly choose between alternative fates in order to give rise to a multitude of distinct cell types. A major challenge in developmental biology is to uncover the dynamics of gene expression and the underlying gene regulatory networks that lead cells to their different fates. Given the complexity of gene regulatory networks, with their large number of components and even larger number of potential interactions between those components, building detailed predictive mathematical models is challenging. The lack of sufficient data requires a large number of assumptions to be made in order to constrain all the parameters in such models (**Karr et al., 2012**).

Our accompanying work on extracting cell states and the sequence of cell state transitions from gene expression data (**Furchtgott et al., 2016**) suggested that following the dynamics of a key set of genes was sufficient to trace these transitions, and in several instances the set of key genes that we discovered were also functionally important for lineage decisions. We asked whether we could similarly determine the suitable parameters to quantitatively describe cell state transitions during

early mammalian germ layer development and build predictive mathematical models of the underlying regulatory network.

Early differentiation of pluripotent mouse embryonic stem (mES) cells, which are derived from the inner cell mass of the peri-implantation stage embryo (see pictorial summary in *Figure 1—figure supplement 1A*), recapitulate various aspects of in vivo germ layer differentiation (*Evans and Kaufman, 1981*; *Keller, 2005*). During this stage, both mES cells and cells in vivo express key pluripotency factors, such as *Nanog*, *Sox2*, *Oct4*, *Klf4*, *Jarid2*, and *Esrrb*, which mutually activate one another to form a pluripotency circuit (*Kim et al., 2008*; *Young, 2011*; *Zhou et al., 2007*). Following implantation, naïve pluripotent ES cells of the inner cell mass downregulate *Klf4* and upregulate *Otx2*, *Dnmt3a*, and *Dnmt3b*, as they transition into 'primed' pluripotent cells found in the epiblast (*Buecker et al., 2014*; *Nichols and Smith, 2009*). Over the next few days of differentiation, TGF-beta signaling factors, with the aid of WNT/beta-catenin signaling, promote and inhibit the differentiation of pluripotent cells into mesendodermal (characterized by genes such as *Brachyury* (*T*), *FoxA2*, *Mixl1* and *Gsc*) and ectodermal (characterized by *Eras*, *Sez6*, *Stmn3*, and *Stmn4*) cell fates, respectively (*Gadue et al., 2006*; *Hart et al., 2002*; *Li et al., 2015*; *Lindsley et al., 2006*; *Tada et al., 2005*; *Watabe and Miyazono, 2009*). Mesendodermal progenitors further differentiate into mesoderm and definitive endoderm progenitors. Mesoderm cells are usually distinguished by the expression of *Gata4* and *Eomes*, and endoderm cells by *Sox17* and *FoxA2*, although in mouse these genes are shared between both lineages, with differences only in their timing and level of expression (*Arnold and Robertson, 2009*; *Kanai-Azuma et al., 2002*; *Kim and Ong, 2012*; *Lumelsky et al., 2001*; *Rojas et al., 2005*). Along the ectodermal lineage, BMP signaling pushes ectodermal cells toward epidermis, while in the absence of BMP signaling, ectodermal cells acquire a neural fate (*Wilson and Hemmati-Brivanlou, 1995*). Epidermal cells are characterized by Keratins, whereas neural cells express *Sox1* and *Pax6* (*Koch and Roop, 2004*; *Pevny et al., 1998*; *Sansom et al., 2009*; *Streit and Stern, 1999*). The cells at the physical border between epidermal and neural cells give rise to neural crest cells (expressing *Sox10*, *Msx2*, *Snai1* and *Slug*) in response to WNT and BMP signaling, which are often described as a fourth germ layer because of the diverse range of tissues to which they give rise (*Gans and Northcutt, 1983*; *Knecht and Bronner-Fraser, 2002*; *Nicole and Chaya, 1991*). Despite the detailed understanding of early embryonic development revealed by decades of work in genetics and developmental biology, a quantitative understanding of how the underlying gene regulatory network leads cells through a series of cell fate decisions has remained elusive.

We use single-cell RNA-seq to determine how gene expression patterns change as mouse embryonic stem cells differentiate into different germ-layer progenitors. We employ a Bayesian framework (*Furchtgott et al., 2016*) to simultaneously infer cell states, the sequence of transitions between these states, and the key sets of genes whose expression patterns provide a parameter space in which the cell states and cell state transitions are inferred. Our computational analysis, together with experimental validation using flow cytometry and live-cell imaging of a new *Otx2* reporter mES cell line, suggest that cells reside in discrete states and rapidly transition from one state to another.

Using the inferred gene expression dynamics and by requiring models to replicate the existence of the observed discrete cell states, we extract probability distributions of the parameters of a model gene regulatory network. Intriguingly, requiring the model to have discrete cell states leads to the prediction that each cell state has a distinct response to perturbations by signals and changing transcription factor expression levels. We experimentally verify three distinct categories of predictions, each testing whether cells exhibit such state-dependent behavior in response to a different type of perturbation. The experimental results conclude that whether (i) *Sox2* overexpression represses *Oct4*, (ii) *Snai1* overexpression represses *Oct4*, and (iii) LIF and BMP promote pluripotency or differentiation into neural crest, all depend on cell state. Finally, we discuss the biological implications of our results.

## Results

### Acquiring single-cell transcriptomics data during early differentiation

We differentiated populations of mES cells by exposing them to one of four combinations of signaling factors and small molecules to perturb key paracrine signaling pathways involved in early

mammalian patterning (*Power and Tam, 1993*; *Tam et al., 2006*): FGF, WNT, and/or TGF-beta signaling for up to five days (*Figure 1A*; see also *Figure 1—figure supplement 1B*, Materials and methods). Although cells in each population were differentiated in a monolayer culture and therefore exposed to nearly uniform conditions, we observed significant heterogeneity in the expression – as measured by immunofluorescence – of various known early germ layer marker genes (such as T, Pax6, Slug, FoxA2, and Gata4) in each population, suggesting a diversity of cell types under the same signaling conditions (*Figure 1B*). Further, undifferentiated pluripotent cells persisted in differentiating populations (*Figure 1—source data 1*, *Figure 2—source data 1*). Therefore, to capture the cell-to-cell variability within differentiating populations, we collected and transcriptionally profiled single cells every 24 hr over the course of five days of differentiation (*Figure 1—source data 1*) using a modified version of CEL-seq (*Hashimshony et al., 2012*). We obtained gene expression data from a total of 288 cells (*Figure 1—figure supplement 1C–J*; Materials and methods) with a median

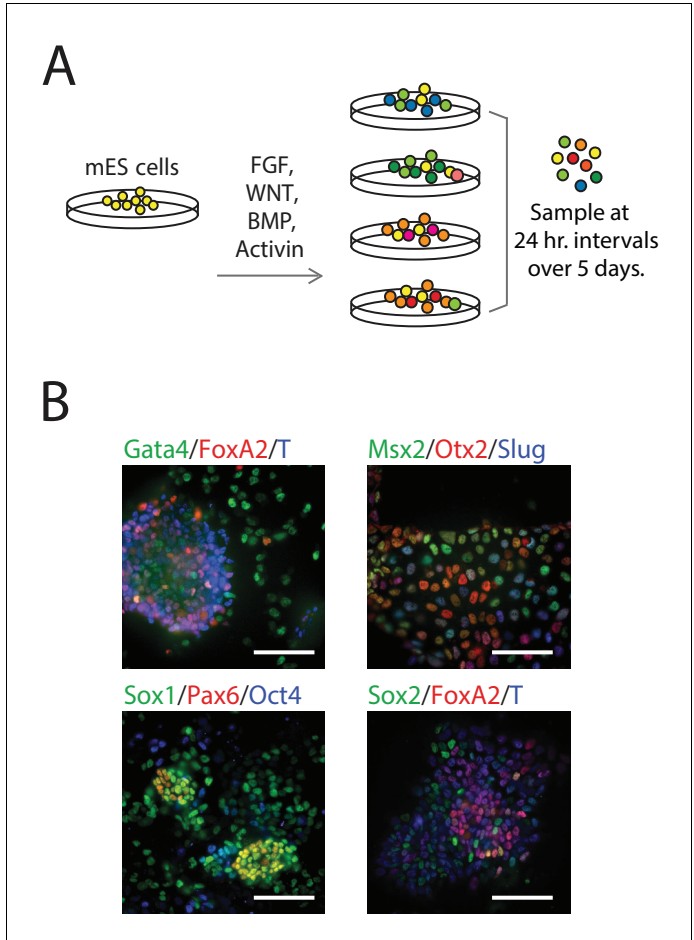

**Figure 1.** Single-Cell Gene Expression Profiling of mESCs during early germ layer differentiation. (**A**) Mouse embryonic stem cells (mESCs) were exposed to various differentiation conditions to perturb FGF, WNT, and TGF-beta signaling for up to five days of differentiation. Single cells, collected every 24 hr during differentiation, were transcriptionally profiled using CEL-Seq. (See also *Figure 1—figure supplement 1B* and *Figure 1—source data 1*). (**B**) Images of immunostained mESCs undergoing differentiation show cell-to-cell variability in their expression of known germ layer marker genes. (Scale bar = 100 μm).

The following source data and figure supplement are available for figure 1:

**Source data 1.** Differentiation conditions and duration of single cells sorted into seven 96-well plates.

**Figure supplement 1.** Quality validation of single-cell RNA-seq data.

of 508,939 mapped reads, 48,475 transcripts and 7032 genes detected per cell. We then randomly subsampled 20,000 reads from each cell to eliminate any technical biases that may have resulted from differences in read numbers across cells (*Figure 1—figure supplement 1K*).

## Bayesian statistical approach discovers appropriate coordinate systems to infer cell states and state transitions

One of the challenges in analyzing single-cell gene expression data is the high dimensionality of the data set and the concomitant sparsity of the data (the number of data points divided by the dimensionality is small) (*Advani and Ganguli, 2016*). Conventional analysis of single-cell gene expression data relies on multi-gene or multi-cell correlation estimates, such as PCA (Seurat) (*Satija et al., 2015*), ICA (Monocle) (*Trapnell et al., 2014*) and WGCNA (*Li et al., 2016*; *Saadatpour et al., 2014*) to reduce the dimensionality of expression data. However, discovering cell types and their lineage relationships using these methods has been challenging (*Furchtgott et al., 2016*).

In the accompanying paper, Furchtgott et al. develop a Bayesian framework that simultaneously infers (i) cell cluster identities of the cells, $\{C\} \equiv \{c_1, c_2, \ldots, c_N\}$,, (ii) the sets of transitions $\{\mathrm{T}\}$ between these clusters, (iii) the key sets of marker genes $\{\alpha_i\}$ that define each cell cluster and (iv) the sets of transition genes $\{\beta_i\}$ that define the transitions between clusters, from single-cell gene expression data $\{g_i\}$, by means of an iterative algorithm to determine the maximum likelihood estimates of these variables (*Furchtgott et al., 2016*).

Here, we employed this Bayesian framework to discover cell types and infer their lineage relationships for early mouse germ layer differentiation. We started by clustering the single-cell gene expression data for the 288 cells into 12 seed clusters $\{c_1^0, c_2^0, \ldots, c_{12}^0\}$ using Seurat (*Satija et al., 2015*) as well as k-means (*Figure 2—figure supplement 2A, B and C*), restricting the analysis to transcription factors (2672 total) because of their functional role in orchestrating global gene expression (*Spitz and Furlong, 2012*). Seurat identifies cell clusters by performing density-based clustering on a two dimensional t-distributed Stochastic Neighbor Embedding (t-SNE) map of the gene expression data (*Van der Maaten and Hinton, 2008*). These clusters $\{C\}^0 = \{c_1^0, c_2^0, \ldots, c_{12}^0\}$, ranging in size from 14 to 47 single cells, served as a seed for the iterative algorithm (described below).

We next considered every possible group of 3 clusters (e.g., $c_1^0$, $c_2^0$ and $c_3^0$) from a total of $^{12}C_3 = 220$ such combinations. For each triplet of clusters, we first determined the probability that each gene $i$ was a marker gene ($\alpha_i = 1$), a transition gene ($\beta_i = 1$) or neither ($\alpha_i, \beta_i = 0$) based on the distribution of their expression patterns in cells of each cluster, where $\{g_i\}$ is the single-cell gene expression data of the $i$−th gene. Marker and transition genes are defined as follows (*Figure 2—figure supplement 1A*, Materials and methods,; *Furchtgott et al., 2016*): (i) A marker gene $i$ ($\alpha_i = 1$) has a distribution of expression levels that is *highest* in one cluster, and well separated from the distribution of its expression levels in the other two clusters. Marker genes distinguish one of the clusters from the other two. (ii) A transition gene j ($\beta_j = 1$) has a distribution of expression levels that is *lowest* in one cluster, and well separated from the distribution of its expression levels in the other two clusters. Each such transition gene establishes relative relationships between the three clusters (*Furchtgott et al., 2016*). (iii) Genes that are neither marker ($\alpha = 0$) nor transition genes ($\beta = 0$) do not follow constraints (i) and (ii) on expression level distributions. Computing the probability of each gene being a marker gene, a transition gene, or neither allowed us to determine the most likely set of transitions $\mathrm{T}$ between each triplet of clusters. Each gene's contribution to the posterior probability $\mathrm{T}$ is weighted by the odds ratio that the gene is a transition gene (*Figure 2—figure supplement 1B*). For example, for clusters $c_1^0$, $c_2^0$ and $c_3^0$, a gene whose expression is lower in $c_2^0$ casts a vote against $c_2^0$ being the intermediate state (i.e., against the transition $\mathrm{T} = c_2^0$, where $c_2^0$ is intermediate, *Figure 2—figure supplement 1B* right) that is weighted by its odds of being a transition gene for those three clusters (*Figure 2—figure supplement 1B*, left). This Bayesian framework led to a summation of these weighted votes to determine the most likely set of transitions between each set of three clusters and concomitantly the most likely marker and transition genes corresponding to these clusters and transitions (*Figure 2—figure supplement 1B*, right).

For the seed cluster set $\{C\}^0$, we determined 179 sets of transitions between clusters and identified 1035 transcription factors that were high probability marker or transition genes for at least one of the identified transitions. For a gene to be defined as a marker or transition gene, we used a

probability cutoff of 0.5. Moreover, we used a probability cutoff of 0.6 for a triplet of clusters to count as a transition event. We next re-clustered the single cells in the gene expression space defined by these 1035 marker or transition genes using Seurat, to obtain a new cluster set $\{C\}^1 = \{c_1^0, c_2^0, \ldots, c_{10}^0\}$ consisting of 10 clusters. In this process, cells changed cluster identities, and certain clusters merged (*Figure 2A*, *Figure 2—figure supplement 1C*).

By iteratively determining the most likely sets of transitions and the most likely marker and transition genes, and by re-clustering the cells within the subspace of these genes, the algorithm converged (i.e., the number of genes of the re-clustering subspace became less than 10% of the total number of transcription factors) upon the most likely set of cell clusters (*Figure 2—source data 1*), the sets of transitions between these cell clusters (*Figure 2—source data 2*), as well as a set of 889 genes categorized as marker or transition genes for at least one set of transitions after five iterations (*Figure 2A*; *Figure 2—figure supplement 2D*).

The final cluster set consists of 9 cell clusters ranging in size between 14 and 57 cells; every cell was mapped to a cluster, and we observed mixing of cells from different experimental conditions to the same cluster as well as cells from the same experimental conditions being assigned to different clusters (*Figure 1—source data 1*; *Figure 2—source data 1*; *Figure 3—figure supplement 1A*). We combined the local sets of transitions between different triplets of clusters (*Figure 2—source data 2*) in order to infer the most parsimonious lineage tree between the clusters (*Figure 2A*) (*Furchtgott et al., 2016*). Importantly, we obtained identical final clusters starting with different seed cluster sets using k-means clustering with the gap statistic, as well as with different threshold probability parameter values for defining transition and marker genes, showing that our results were robust to the choice of seed clusters, threshold probability value and clustering method (*Figure 2— figure supplement 2A, B and C*; *Figure 2—figure supplement 3A and B*; Materials and methods). The cluster identities as well as their lineage relationships were unchanged when the analysis was repeated with a subset of cells; in which either an entire cluster was removed or a random set of half (144) cells were removed (*Figure 2—figure supplement 3C and D*). Further, we found that the clustering configuration does not change depending on whether the analysis is restricted to only transcription factors or includes all genes (*Figure 2—figure supplement 3E*). However, using all genes resulted in greater error rates along the topology of the inferred lineage tree compared to when only transcription factors were used (*Furchtgott et al., 2016*; *Figure 2—figure supplement 3F*).

The inferred lineage relationships between the final clusters could be visualized in the subspace of inferred marker and transition genes. We illustrate this first for the three clusters $C_1$, $C_2$, and $C_3$. We identified three classes of marker genes, each consisting of high-probability marker genes specific to one of the three clusters (*Figure 2B*). Each gene class is denoted by its highest probability member gene in curly brackets (e.g., {*Otx2*}). When the cell-cell Pearson correlation matrix between all 288 cells was determined using the 889 genes used for the final iteration of clustering and lineage determination, the matrix showed a barely detectable structure of nine blocks (with very low contrast) along the diagonal with marginally higher correlation levels, each corresponding to a cell cluster (*Figure 2D*). As expected, the low level of contrast observed in *Figure 2D* improves dramatically when the same correlation measures are taken across cells in a triplet, using marker or transition genes for this triplet; illustrating the locally defined nature of marker and transition genes (*Figure 2B*, right; *Figure 2C*, right; *Figure 2—figure supplement 2E*). The same matrix computed using high-probability marker genes for clusters $C_1$, $C_2$, and $C_3$ (*Figure 2B*, left) showed three distinct blocks of high correlation along the diagonal, each corresponding to a different cluster (*Figure 2B*, right). Similarly, when the cell-cell correlations were measured using the two classes of inferred transition genes (*Figure 2C*, left), each consisting of high-probability transition genes present in $C_1$ and downregulated either in $C_2$ or in $C_3$, the correlation matrix showed intermediate correlation levels between $C_1$ and either $C_2$ or $C_3$, and low correlation levels between $C_2$ and $C_3$ (*Figure 2C*, right). The distribution functions of the expression levels of these transition genes in each of the three different clusters ($C_1$, $C_2$ and $C_3$) led to the inference that clusters $C_2$ and $C_3$ are connected via cluster $C_1$ with a probability of 0.83 (*Figure 2E*, *Figure 2—figure supplement 1A and B*).

We visualized the gene expression changes that characterize transitions from one cell cluster to another by plotting the cells in $C_1$, $C_2$ and $C_3$ in a three-dimensional gene expression subspace (*Figure 2E*), using as axes the mean normalized expression levels of the two transition gene classes

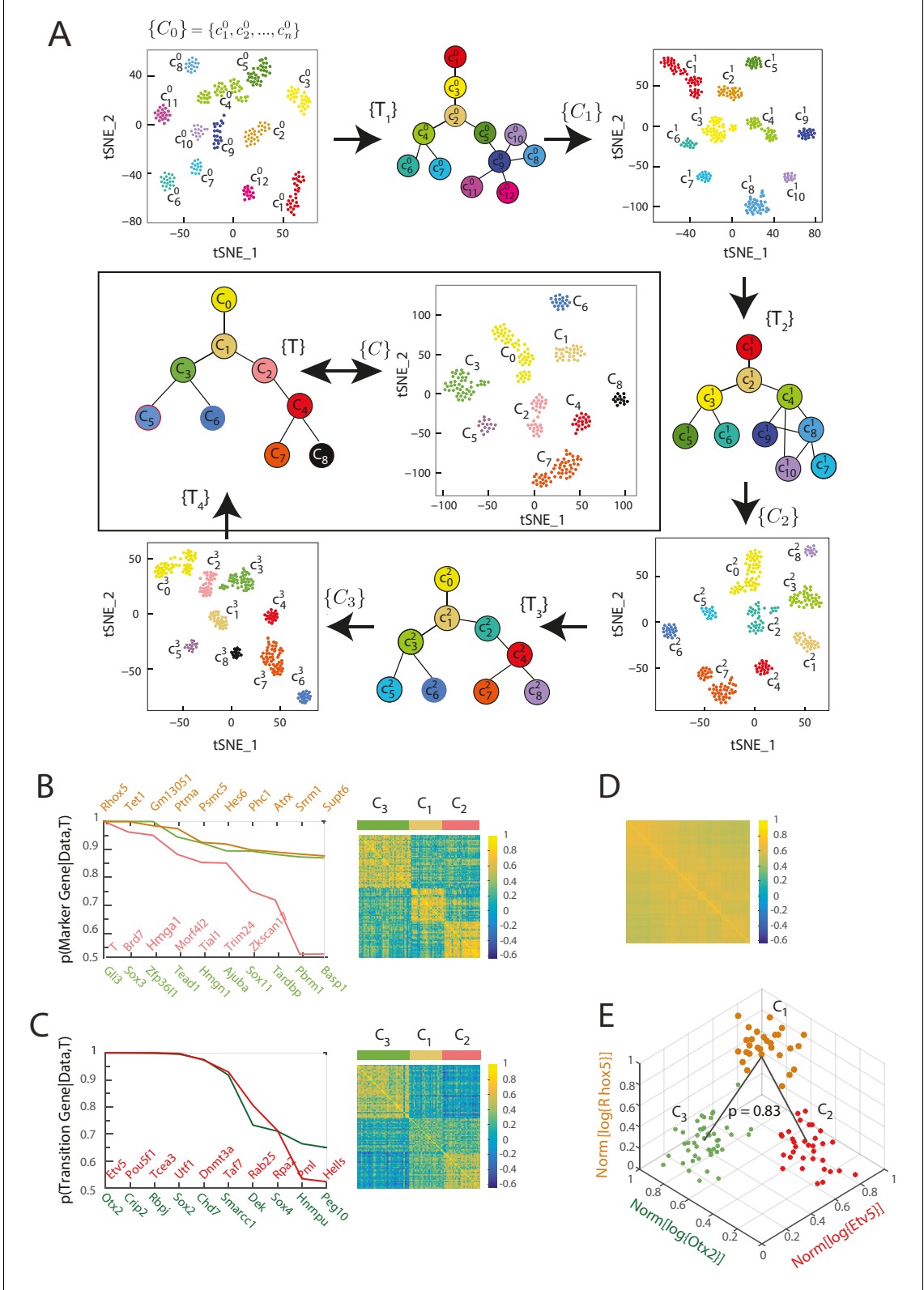

**Figure 2.** Iterative Bayesian algorithm converges upon a set of cell clusters and local transitions that together define a multi-potent lineage tree. (**A**) Iterative determination of the most likely sets of transitions $\{T\}$ and re-clustering of cells in the resulting subspace of transition and marker genes, starting from a seed set of cluster identities $\{C\}^0$. With each iteration, the cluster identities as well as the total number of clusters change, as shown by the Seurat t-SNE maps (each dot represents a cell, colored based on its cluster identity). The inferred sets of transitions between clusters at each

*Figure 2 continued on next page*

*Figure 2 continued*

iteration are represented as a lineage tree (each circle represents a cell cluster). After five iterations, the algorithm converged upon a set of 9 clusters (shown in box). (See also *Figure 2—figure supplement 2*). (B) Left: Top ten genes (x-axis) with highest probability of being marker genes for clusters $C_1$ (yellow), $C_2$ (light red) and $C_3$ (light green) plotted against their probability of being marker genes. Right: Cell-cell correlation matrix computed using these 30 marker genes for the 108 cells belonging to clusters $C_1$, $C_2$ and $C_3$ shows three clear blocks of high correlation along the diagonal. (C) Left: Top ten genes (x-axis) with highest probability of being transitioned genes for clusters $C_1$, $C_2$ and $C_3$, plotted against their probability of being transitioned genes (y-axis). The transition genes belong to one of two classes, those that show high expression in cells belonging to $C_1$ and $C_2$ but low expression in $C_3$ (red), and those expressed at high levels in cells in clusters $C_1$ and $C_3$ but low levels in $C_2$ (green). The cell-cell correlation matrix computed using these 20 transition genes shows that the 29 cells belonging to cluster $C_1$ have intermediate levels of correlation with cells in both $C_2$ and $C_3$, whereas the 46 cells in $C_2$ show low correlation levels with the 33 cells in $C_3$. (D) The global cell-cell correlation matrix computed for all 288 cells using the 889 genes used for the final iteration of clustering shows a barely detectable structure. (E) The inferred clusters and their lineage relationships can be represented in a three-dimensional coordinate system where the x- and y- axes are the normalized log expression level of the two classes of transition genes (genes in *Figure 2B*, left) and the z-axis measures the normalized log expression level of the marker genes for cluster $C_1$ (*Figure 2A* left in yellow). Each dot represents a single cell, and cells are colored based on their cluster identity.

The following source data and figure supplements are available for figure 2:

**Source data 1.** Plate and well id's of cells belonging to each cluster.
**Source data 2.** Triplet probabilities of final tree.
**Figure supplement 1.** Diagram of Bayesian framework for inferring sequence of transitions for triplets.
**Figure supplement 2.** Iterative clustering and lineage determination is robust to clustering method.
**Figure supplement 3.** Iterative clustering and lineage determination is robust to changes in parameters.

down-regulated in $C_2$ or $C_3$ (in red and green in *Figure 2C*) and of the marker gene class specific to $C_1$ (*Figure 2B* in orange). These axes constitute a low-dimensional coordinate system for the inferred set of transitions between $C_1$, $C_2$ and $C_3$.

Similarly, the inferred transitions across all sets of three clusters (*Figure 2—source data 2*) together form a lineage tree (*Figure 3A*) that spans all nine identified cell clusters, which can be visualized in gene expression space through a series of local transition and marker gene classes (*Figure 3C*; *Figure 3—source data 1*). We next investigated the gene expression variability among cells within each cluster by performing principal component analysis (PCA) on the transcription factor gene expression for cells within each cluster. Importantly, we found that for all clusters, no principal component is statistically significant (compared to randomizations of the data; *Figure 3B*, *Figure 3—figure supplement 1B*), validating that within each inferred cluster, the cells have the same identity within the resolution of our data.

The inferred dynamics of differentiation can therefore be visualized in a low-dimensional subspace of gene expression, showing that differentiation occurs through a sequence of discrete cell state transitions.

## Correspondence of cell states discovered ab initio from single-cell data to known in vivo cell types

Inspection of the genes that make up the local transition and marker gene classes (*Figure 3C*; *Figure 3—source data 1*) allowed us to match clusters to embryonic cell types found in vivo that show similar gene expression.

Cluster $C_0$ is characterized by the high expression of pluripotency genes *Oct4*, *Sox2*, *Sall1*, *Etv5*, *Jarid2*, *Esrrb*, *Klf4* and *Klf5*, whereas cluster $C_1$ has lower *Jarid2*, *Esrrb*, *Klf4* and *Klf5*, and higher *Otx2*, *Bptf*, *Cbx1* and *Dnmt3a/b* expression compared to cluster $C_0$, suggesting that clusters $C_0$ and $C_1$ correspond to naïve ES and primed epiblast pluripotent cell types, respectively (*Borgel et al., 2010*; *Goller et al., 2008*; *Kim et al., 2001*; *Nichols and Smith, 2009*; *Tesar et al., 2007*; *Zhou et al., 2007*).

Clusters $C_2$ and $C_3$, which branch out from $C_1$, show differential expression of pluripotency genes relative to $C_1$; *Bptf* and *Cbx1* are downregulated in both $C_2$ and $C_3$, *Oct4*, *Etv5* and *Dnmt3a* are

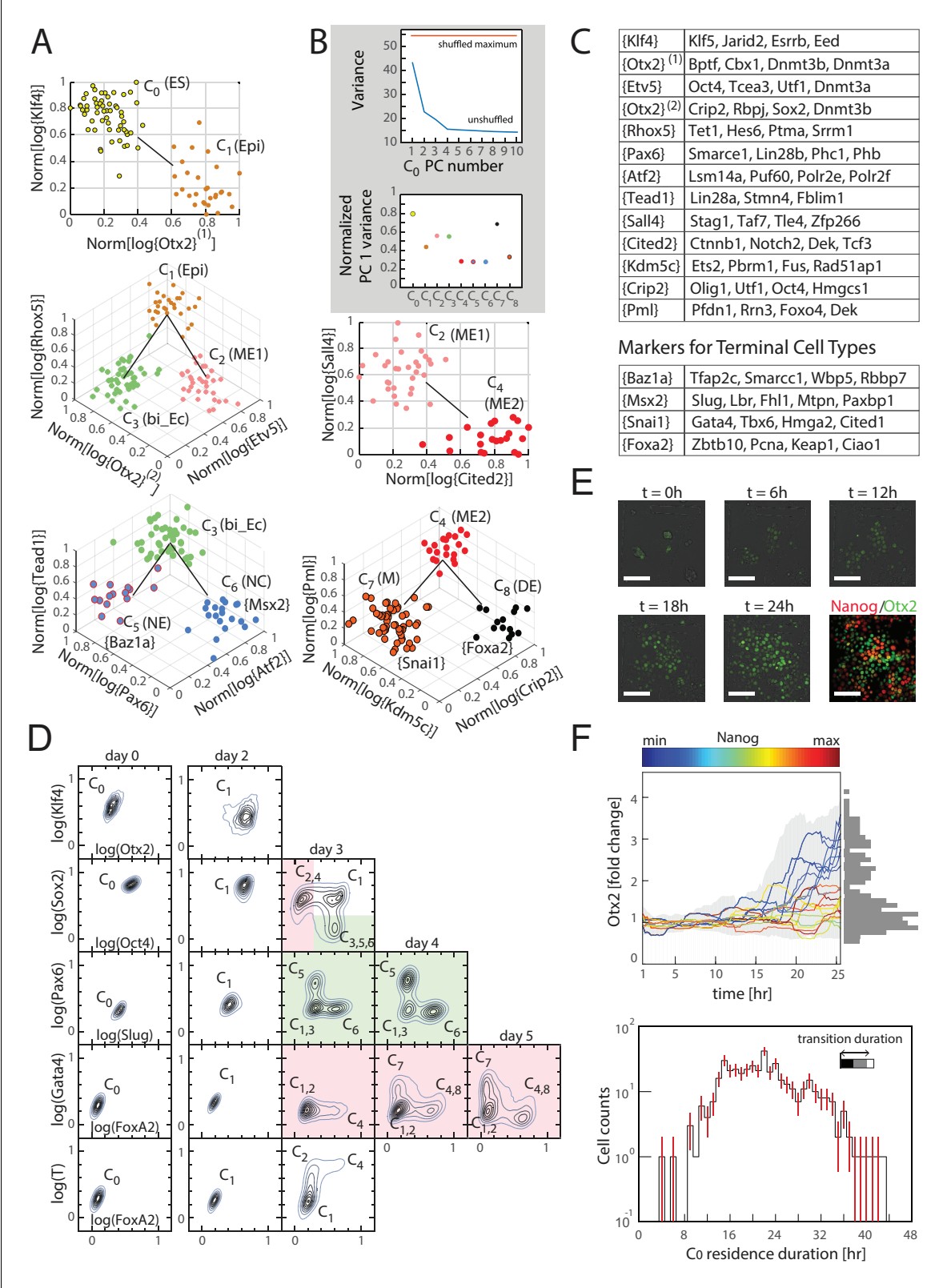

**C**

| {Klf4} | Klf5, Jarid2, Esrrb, Eed |
|---|---|
| {Otx2} [1] | Bptf, Cbx1, Dnmt3b, Dnmt3a |
| {Etv5} | Oct4, Tcea3, Utf1, Dnmt3a |
| {Otx2} [2] | Crip2, Rbpj, Sox2, Dnmt3b |
| {Rhox5} | Tet1, Hes6, Ptma, Srrm1 |
| {Pax6} | Smarce1, Lin28b, Phc1, Phb |
| {Atf2} | Lsm14a, Puf60, Polr2e, Polr2f |
| {Tead1} | Lin28a, Stmn4, Fblim1 |
| {Sall4} | Stag1, Taf7, Tle4, Zfp266 |
| {Cited2} | Ctnnb1, Notch2, Dek, Tcf3 |
| {Kdm5c} | Ets2, Pbrm1, Fus, Rad51ap1 |
| {Crip2} | Olig1, Utf1, Oct4, Hmgcs1 |
| {Pml} | Pfdn1, Rrn3, Foxo4, Dek |

**Markers for Terminal Cell Types**

| {Baz1a} | Tfap2c, Smarcc1, Wbp5, Rbbp7 |
|---|---|
| {Msx2} | Slug, Lbr, Fhl1, Mtpn, Paxbp1 |
| {Snai1} | Gata4, Tbx6, Hmga2, Cited1 |
| {Foxa2} | Zbtb10, Pcna, Keap1, Ciao1 |

**Figure 3.** Cells transition from one discrete state to another during differentiation. (**A**) Computationally inferred cell clusters and sequence of transitions are shown in the appropriate subspace of gene expression. Each dot represents a single cell, and cells are colored based on their cluster identity. For a linear transition sequence of cell states (such as from $C_0$ to $C_1$), the transitions are represented in a two dimensional plot with the axes defined by the normalized mean log of the unique reads of genes that are most differentially regulated in the two states, while for lineage bifurcations between

*Figure 3 continued on next page*

Figure 3 continued

alternative daughter cell states, the plots are shown in three dimensions, where the x and y axes are normalized mean log unique reads of the associated set of transition genes, and the z axes are the normalized mean log unique reads of the marker genes associated with the inferred progenitor state. Labeled in parenthesis next to each cluster are the abbreviated names of the putative corresponding cell types found in vivo (Epi: epiblast; bi_Ec: bi-potent ectoderm; ME: mesendoderm; NE: neural ectoderm; NC: neural crest; M: mesoderm; DE: definitive endoderm). (B) Top: Plot of the variances of the first ten principal components of the gene expression of cells in cluster $C_0$. The red line is the maximum principal component variance over 1000 randomizations of the data, showing that no principal component is statistically significant. Bottom: variances of the first principal component of each cluster, normalized by the maximum principal component variance of the randomized gene expression data for the corresponding cluster. (C) A list of high probability genes that belong to the various marker and transition gene classes that define the axes of the plots in *Figure 3A*, each represented by one gene in curly brackets. The curly brackets contain the gene name with the highest probability for that class, and other high probability genes (as in *Figure 2A and B*) are listed in the table. While some of the genes are used only once, others such as *Otx2* and *Oct4* are repeatedly reused in different subspaces to describe the transition. (D) Flow cytometry analysis of cell populations sampled every 24 hr during differentiation and immunostained for nine genes (two shown at a time for each density contour plot): Klf4, Otx2, Oct4, Sox2, Slug, Pax6, FoxA2, Gata4 (each taken from a different gene class shown in *Figure 3C*), and T recapitulate the predicted structure and temporal ordering of transitions through discrete cell states. Axes represent the log of gene expression, normalized by the range between the minimum and maximum across each gene. Plots in pink and green represent $C_2$ and $C_3$ lineages following the split from $C_1$, respectively. (E) Live cell microscopy of *Otx2* reporter (mCitrine) cell line to infer the dynamics of cell state transition from $C_0$ to $C_1$. Sample images (shown) at t = 0, 6, 12, 18, and 24 hr of differentiation. Cells were terminated at approximately 25 hr into differentiation and immunostained for Nanog (ES marker gene, *Figure 2—figure supplement 2A*), which shows an anti-correlation between Otx2 and Nanog expression levels. (Scale bar = 100 μm) (F) Top: Time series (x-axis) traces of single-cell Otx2 (y-axis) expression dynamics taken every 15 min show that the duration of transition from Otx2-low ($C_0$) to Otx2-high ($C_1$) is approximately 4 hr, which is well within the time frame of one cell cycle (~10 hr). The end-point (t = 25 hr) Otx2 levels show a clear separation between high and low (histogram of ~200 cells shown to the right in gray), indicating that some cells have made the transition from $C_0$ to $C_1$ while others not. Each trace is colored by its relative end-point Nanog immunofluorescence intensity level. Otx2 levels are normalized by the mean level at t = 0. Bottom: Histogram (y-axis = log (cell count)) of residence durations of ~400 cells in the Otx2-low $C_0$ state, showing that transition times vary across multiple cell cycle lengths (time lapse length = 48 hr). Inset bar shows mean as well as upper (white) and lower quartiles of the transition durations of cells.

The following source data and figure supplement are available for figure 3:

**Source data 1.** Probabilities of membership in marker and transition gene classes in final tree.

**Figure supplement 1.** Validation of inferred cell types and lineage relationships.

---

downregulated in cluster $C_3$ but maintained in $C_2$, and *Sox2*, *Otx2* and *Dnmt3b* are downregulated in cluster $C_2$ but maintained in cluster $C_3$. Cluster $C_2$ is further characterized by a high expression level of primitive streak markers *Mixl1* and *T* (*Hart et al., 2002*; *Tada et al., 2005*), whereas cluster $C_3$ is characterized by *Sez6*, *Stmn3* and *Stmn4*, which have recently been shown to characterize the previously elusive mammalian bi-potent ectoderm progenitor population (*Li et al., 2015*). Together, these patterns strongly suggest that clusters $C_2$ and $C_3$ represent mesendoderm and bi-potent ectoderm progenitor cell types, respectively.

The bi-potent ectoderm progenitor-like cluster $C_3$ is then followed by a lineage split into clusters $C_5$ and $C_6$. While *Stmn4* is downregulated in both $C_5$ and $C_6$ compared to $C_3$, *Sez6* is downregulated in only $C_5$, and *Stmn3* as well as neural progenitor marker Pax6 are downregulated in $C_6$ but maintained in $C_5$. Cluster $C_5$ is further characterized by *Smarce1* and *Zic2*, and cluster $C_6$ by *Slug* and *Msx2*, suggesting that $C_5$ and $C_6$ may be related to neural progenitor and neural crest cells, respectively (*Brown and Brown, 2009*; *Nicole and Chaya, 1991*; *Vogel-Ciernia and Wood, 2014*).

Cluster $C_4$, although similar in its expression level of *Mixl1* and *T* to cluster $C_2$, shows higher expression of other primitive streak genes such as *FoxA2* and *Tcf3* (*Merrill et al., 2004*) and lower expression of *Etv5*. Cluster $C_4$ is then followed by a bifurcation between clusters $C_7$ and $C_8$. Cluster $C_7$ shows high expression levels of *Gata4* and *Snai1*, indicative of its relation to mesoderm, and cluster $C_8$ is characterized by high *FoxA2* compared to clusters $C_4$ and $C_8$, suggestive of its relation to definitive endoderm (*Kim and Ong, 2012*; *Rojas et al., 2005*). We predict that cluster $C_4$ represents a primed bi-potent mesendoderm cell type relative to cluster $C_2$ (*Nakanishi et al., 2009*).

Together, these results suggest that the cell clusters and sets of transitions computationally inferred from single-cell transcriptomics data correspond to known in vivo cell types and their lineage relationships.

## Differentiation occurs through a series of discrete cell state transitions

The fact that gene expression in each cell cluster does not vary significantly – as measured by the relative sizes of the largest eigenvalues of the PC components of the gene expression data (or percent variance explained thereby) versus that of the same data randomly shuffled (*Figure 3B*; *Figure 3—figure supplement 1B*) – allows for genes to be sorted into a few gene classes that show highly correlated expression patterns across clusters (*Figure 3C*). This suggests that one can validate the inferred sequence of cell state transitions and its gene expression dynamics by measuring the expression of one gene from each class in differentiating cells over time.

In order to confirm the gene expression dynamics over the inferred sequence of cell state transitions, we assessed populations of cells for their expression levels of key transition and marker genes (each taken from a different gene class) via immunostaining and flow cytometry. We sampled mES cell populations every 24 hr during differentiation and immunostained each for Klf4, Otx2, Oct4, Sox2, Pax6, Slug, FoxA2, Gata4 and T. (Although $T$ is not assigned to a specific gene class, it is highly expressed in the mesendoderm-like states $C_2$ and $C_4$, and it thus allows us to distinguish $C_2$ from the earlier epiblast-like state $C_1$.) The flow cytometry density contour plots shown (*Figure 3D*) are characterized by high-density peaks which are separated from one another by regions of low density, mirroring the discreteness of the cell states inferred from single-cell transcriptomics data. The relative locations of these high-density peaks and the time at which they appear and disappear recapitulate the inferred gene expression dynamics of the cell state transitions of the lineage tree.

During the first two days of differentiation, all cell populations downregulated Klf4 and upregulated Otx2, as shown in the first row of density contour plots in *Figure 3D*. This is consistent with the first observed state transition in our inferred lineage tree from the naïve ES $C_0$ state to the primed epiblast-like state $C_1$. On day three of differentiation (third column of plots in *Figure 3D*), Sox2 and Oct4 are asymmetrically downregulated relative to the preceding population, as is seen in mesendoderm-like state $C_2$ and bi-potent ectoderm-like state $C_3$ relative to the epiblast-like state $C_1$. Sox2-high, Oct4-low cells on day three are either high for Pax6 or for Slug, consistent with comparisons between the neural ectoderm-like state $C_5$ and neural crest-like $C_6$. On day four, the Pax6-high and Slug-high populations become proportionally larger as the Pax6/Slug-low population shrinks, supporting the inferred temporal ordering that $C_5$ and $C_6$ arise from the bi-potent ectoderm-like state $C_3$. Oct4-high, Sox2-low cells on day three of differentiation are high for T, but show two discrete levels of FoxA2, mirroring the difference between the two mesendoderm-like states $C_2$ (FoxA2-low) and $C_4$ (FoxA2-high). Further, we found that Etv5, a gene whose expression dynamics had hitherto not been implicated with early mesendodermal differentiation in mammals, was significantly downregulated from $C_2$ to $C_4$, as predicted from the single-cell gene expression data (*Figure 3—figure supplement 1C*). Finally, at days four and five, we observe FoxA2-high, Gata4-low and FoxA2-low, Gata4-high cell populations, which correspond to the primed mesendoderm and definitive endoderm-like states $C_4$ and $C_8$ and the mesoderm-like state $C_7$, respectively. We thus confirmed that differentiating cell populations recapitulate the gene expression dynamics of cell state transitions inferred from single-cell data (*Figure 3A*).

The observation that the majority of randomly sampled cells are found to belong to one of nine discrete cell states (both transcriptionally and at the protein level) suggests that cell state transitions occur within a relatively short timeframe compared to the amount of time cells spend within each state. We tested this hypothesis on the first cell state transition from the naïve ES $C_0$ state to the primed epiblast-like state $C_1$ (*Figure 3A*). To do so, we generated an *Otx2*-mCitrine fusion protein reporter mES cell line (Materials and methods) and observed the single-cell-resolution dynamics of *Otx2* expression for up to two days (*Figure 3E and F*).

In agreement with our hypothesis, we observed that Otx2 levels, at the end of 24 hr of differentiation, show a bimodal distribution (*Figure 3F*, top), and cells tend to occupy either an Otx2-low state (corresponding to ES state $C_0$) or an Otx2-high state (corresponding to epiblast-like state $C_1$). We find that cells transition from an Otx2-low to an Otx2-high state well within the duration of a single cell cycle (mean transition duration of 4.52 hr compared to the cell-cycle length of approximately 10 hr). In contrast, cells tend to stay in either Otx2-low or -high states for up to multiple cell cycles, with a large amount of cell-to-cell variability in the residence duration (*Figure 3F*, bottom). Together with our results from the analysis of single-cell transcriptomics data, these observations show that

cells reside in discrete states in gene expression space and correspondingly undergo abrupt state transitions.

## A probabilistic model that replicates the observed discrete cell states predicts state-dependent interpretation of perturbations

Our analysis of single-cell gene expression data suggested a lineage tree composed of discrete cell states, and identified genes associated with individual cell states and transitions between them. While we predict the existence of discrete cell states based on their gene expression pattern, finding unique physiological properties that can define and distinguish their existence functionally would lend even greater support to this prediction. We therefore next sought to find properties of cell states that distinguished them functionally from one another. In order to do so, we built a predictive and testable quantitative model of the underlying gene regulatory network based on the expression patterns of the marker and transition genes.

From the 889 genes that were categorized as either marker or transition genes for all the high probability triplets, we first chose genes involved only in the triplets that fall directly along the inferred lineage tree. That is, we removed genes that were categorized as transition or marker genes for triplets consisting of 'indirect' lineage relationships, where at least one cell state is skipped between two cell states connected through the lineage tree. For instance, we did not consider the genes categorized as marker or transition genes only in the triplet $C_0$, $C_1$ and $C_5$, because $C_3$ is skipped between $C_1$ and $C_5$.

Since some transition genes inferred from our Bayesian analysis are re-used to infer multiple local state transitions (*Figure 3C* ,e.g., *Oct4*, *Otx2*), we classified transcription factors based on their distinct binarized patterns of expression across all nine cell states, with genes showing the same patterns belonging to the same gene module (Materials and methods, *Figure 4—figure supplement 1A*, *Figure 4—source data 1*). Hence, we categorized the 321 marker and transition genes involving 'direct' triplets along the tree into 26 gene modules, each of which showed distinct patterns of expression across the cell states. Further, because our goal was to test whether different cell states were functionally distinct (i.e., respond differently to the same signals and gene expression changes), we also noted the expression pattern of signaling factor genes belonging to FGF, WNT, LIF and BMP signaling pathways along the lineage tree (*Figure 3A*). These signaling factor genes constituting each of these modules were selected based on GO categories, leading to a total of 29 gene modules (Materials and methods). We denote each gene module by a representative gene in square brackets; for example, the gene module that uniquely characterizes the ES state $C_0$ is denoted as [*Klf4*] (*Figure 4—source data 1* and *2*).

Owing to the large number of gene modules, and consequently even larger number of potential interactions between these modules, even the simplest mathematical model would consist of hundreds of parameters. However, for most of these parameters, direct experimental measurements are not available. In order to overcome this challenge, we exploited recent developments based on renormalization group approaches to determine which parameters are relevant for the observed data (*Machta et al., 2013*). We adapted the seminal model of artificial neural networks, known as the Hopfield model (*Fard et al., 2016*; *Hopfield, 1984*; *Maetschke and Ragan, 2014*), to construct an effective gene regulatory network between the 29 gene modules. By construction, we required that this mathematical model produces the nine cell states seen in *Figure 3A*. We considered a network that contains direct interactions, in which each module $j$ exerts a drive on module $i$, which is equal to an interaction strength $J_{ij}$ (positive or negative) multiplied by the concentration of module $j$. The total drive on module $i$ is the sum of the drives from the different modules. Given our observation of discrete cell states, we further considered that the total drive on module $i$ affects expression in a highly non-linear manner, with high gene expression for drives that exceed a critical drive $\phi_0$, and low gene expression otherwise (*Figure 4—figure supplement 1B*). For simplicity, we assumed that the expression of every gene module exhibits a non-linear, step-function response, when subjected to the same drive; thereby reducing the number of parameters of the model. Indeed there are numerous genes that manifest sigmoidal-like response in expression, in the presence of internal and external stimuli (*Lebrecht et al., 2005*; *Segal and Widom, 2009*). Thus the effective dynamics of expression levels $m_i$ of each module $i$ are given by the non-linear equation:

$$\frac{dm_i}{dt} = H\left(\sum_j J_{ij}m_j - \phi_0\right) - \frac{m_i}{\tau_i}$$

where $H$ is the Heaviside step function and $\tau_i$ is the effective lifetime of module $i$ (Materials and methods).

We determined the set of interactions $J_{ij}$ that are consistent with the observed cell states ($C_0$-$C_8$, Figure 3A) being stable fixed points of the network. If state $\vec{m}^\alpha = \{m_1^\alpha, \ldots, m_{29}^\alpha\}$ with expression level $m_i^\alpha$ in module $i$ is a stable fixed point of the network, then the interactions $J_{ij}$ must be such that the total drive on each module that is expressed in $\vec{m}^\alpha$ is greater than the critical drive, and the total drive on each module that is not expressed in $\vec{m}^\alpha$ is less than the critical drive:

$$m_i^\alpha = 1 \Rightarrow \sum_j J_{ij}m_j^\alpha \geq \phi_0$$

$$m_i^\alpha = 0 \Rightarrow \sum_j J_{ij}m_j^\alpha < \phi_0$$

Thus, for each stable state, we have 29 constraints on the possible values of $J_{ij}$, one for each module. Given that we have nine cell states, there are 29*9 = 261 inequalities that constrain the values of the $29^2$ = 841 different parameters, $J_{ij}$. The problem is therefore underdetermined even for our simplified model of the underlying network, and there are an infinite number of solutions that would allow for the observed cell states to be stable.

By using a linear programming method to obtain an ensemble of 10,000 sets of $J_{ij}$ interactions (Materials and methods), each satisfying the constraint that all nine cell states are stable fixed points, we estimated the probability distribution for the 841 parameters of the model (Figure 4A and Figure 4—figure supplement 2), giving us a probabilistic model of the underlying network. We further assumed that all the possible 10,000 sets of $J_{ij}$ interactions that reproduced the nine stable cell states were equally likely, since we did not have any experimental evidence to distinguish between them.

We used this probabilistic model to make testable predictions as to how different cell states respond to perturbations: to see if different cell states are defined not only by their distinct transcriptional profiles, but also functionally distinct in their phenotypic responses to the same perturbations. There are a vast number of testable predictions that one could extract from our gene regulatory network model. However, given the low throughput nature of perturbation experiments, we selected three distinct probabilistic predictions, each probing different aspects of the model gene regulatory network.

First, we considered changes in the effective interaction between two gene modules as a function of cell state (i.e., how the expression level changes of one gene module affect the expression of another gene module differs across cell states due to the difference sets of gene modules present in each state). To this end, we looked at two classes of gene module pairs: (i) gene modules that are co-expressed in two mother-daughter cell states and (ii) gene modules that are never co-expressed in any cell state.

Gene modules [Sox2] and [Oct4] are highly expressed in both the ES cluster $C_0$ and the epiblast-like $C_1$ cluster, after which they are asymmetrically downregulated in the mesendoderm-like $C_2$ and ectoderm-like $C_3$. We find that for 67.5% of the 10,000 sampled solutions, [Sox2] and [Oct4] have mutually inhibitory interactions (i.e., negative coupling constants). Although both [Sox2] and [Oct4] are present together in the $C_0$ and $C_1$ states, their effective interactions are altered in different ways in each cell state by the presence of other gene modules. As cells transition from state $C_0$ to $C_1$, they downregulate gene modules [Klf4], [Atf2], [Apex1] and [Ets2], and upregulate [Hes6] and [Otx2], among others (Figure 4B and C), leading to changes in the effective interaction strength between [Sox2] and [Oct4]. By incrementally increasing [Sox2] levels relative to its base value and assessing the fraction of models that show [Oct4] downregulation, we found that [Oct4] levels are predicted to be more stable to [Sox2] overexpression in state $C_0$ than in $C_1$ (Figure 4D), thus distinguishing $C_0$ and $C_1$ functionally (Geula et al., 2015).

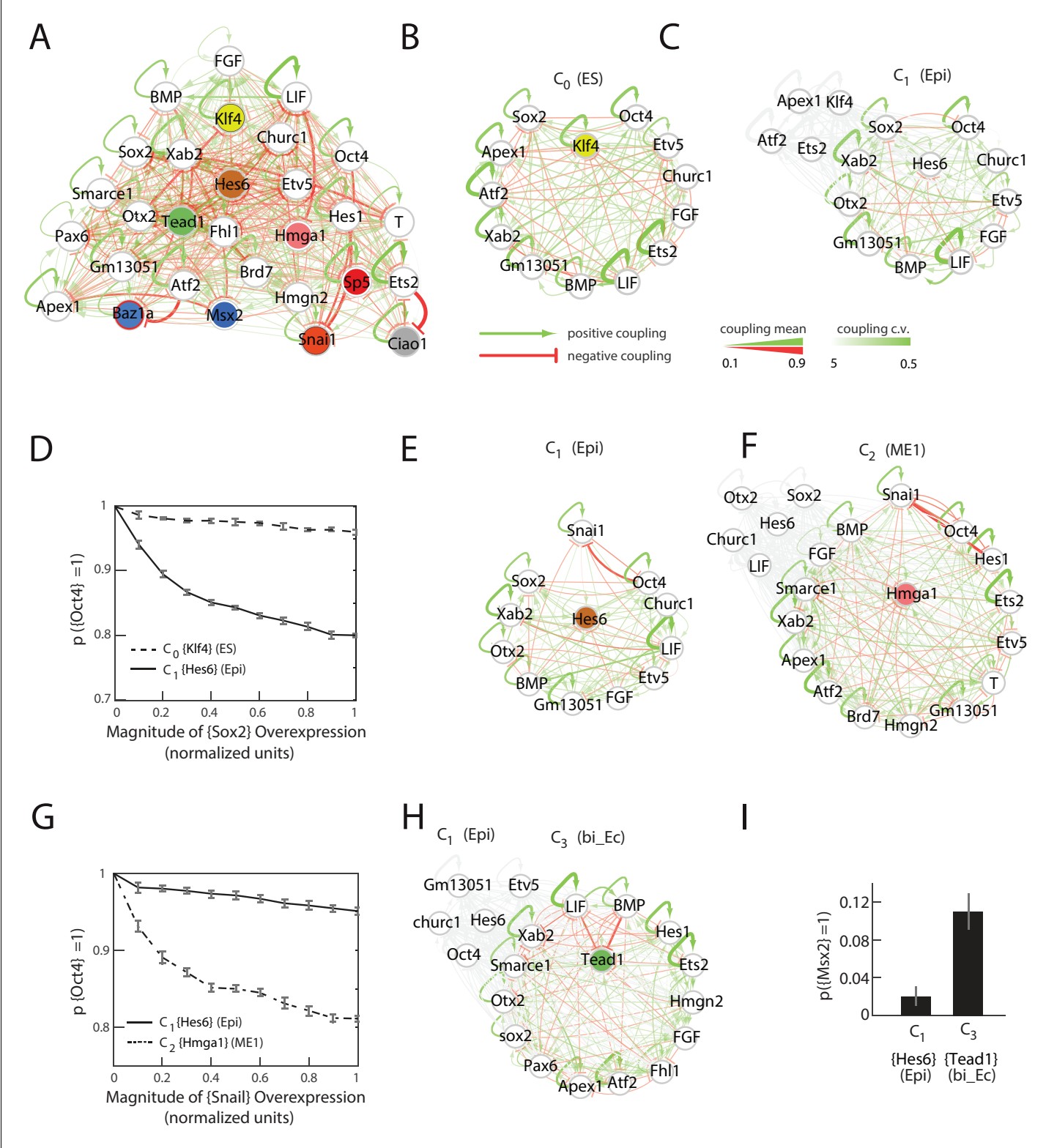

**Figure 4.** Quantitative modeling of the network underlying germ layer differentiation. (**A**) The inferred gene regulatory network from 10,000 sampled solutions that stabilize each of the nine cell states. Each circle represents a gene module. Mean positive and negative interactions between the modules are shown in red and green, respectively, and their thickness and transparency are proportional to the absolute magnitude of the mean and the coefficient of variation (c.v.), respectively. The colored circles represent the gene modules expressed uniquely in only one of the cell states (color code matched with *Figure 3A* for each state). (**B, C**) Subsets of the network consisting of gene modules that are expressed in (and stabilize) the naïve

*Figure 4 continued on next page*

*Figure 4 continued*

ES $C_0$ state (**B**) and epiblast-like $C_1$ (**C**) state. As cells transition from $C_0$ to $C_1$, expression of [*Klf4*], [*Apex1*], [*Ets2*], [*Atf2*] modules is downregulated (shown in gray) while [*Hes6*] and [*Otx2*] modules are upregulated, leading to changes in the effective interactions between gene modules that are common to both $C_0$ and $C_1$ states, such as [*Sox2*] and [*Oct4*]. (**D**) [*Sox2*] overexpression (x-axis) plotted against the probability of [*Oct4*] downregulation (y-axis) computed over 10,000 models (Materials and methods). In the $C_1$ state (solid line), [*Oct4*] is downregulated in an increasing fraction of models following [*Sox2*] overexpression, while in $C_0$, [*Oct4*] is stable in ~96% of the models (dotted line). In order to obtain the error bars for this and subsequent predictions, we randomly sampled three subsets of 3333 from the 10,000 models. For each set we computed the mean and standard error of the proportion of models that show downregulation of *Oct4* in response to *Sox2* overexpression. (**E, F**) Subsets of the model consisting of gene modules that are expressed in the epiblast-like $C_1$ (**E**) and mesendoderm-like $C_2$ (**F**) states, and their interactions with [*Snai1*], which is not normally expressed in $C_1$ or $C_2$. As cells transition from the $C_1$ to $C_2$ state, [*Hes6*], [*Sox2*], [*Otx2*], [*Churc1*] are downregulated (shown in gray), while [*Hmga1*], [*T*], [*Atf2*], [*Hes1*], [*Ets2*], [*Apex1*], [*Brd7*], [*Hmgn2*] and [*Smarce1*] are upregulated, leading to changes in the effective interactions between [*Snai1*] and modules that are common to both $C_1$ and $C_2$, such as [*Oct4*]. (**G**) The probability of [*Oct4*] being downregulated (y-axis) as a function of [*Snai1*] overexpression (x-axis). In the $C_1$ state (solid line), the over expression of [*Snai1*] has no effect on [*Oct4*] levels in ~94.5% of the 10,000 models whereas in the $C_2$ state (dotted line), the overexpression of [*Snai1*] leads to [*Oct4*] downregulation in up to 19% of the models. (**H**) The $C_3$ state shows a downregulation of [*Oct4*] and [BMP], and upregulation of [*Tead1*], [*Apex1*], [*Pax6*], [*Smarce1*], [*Ets2*], [*Atf2*], [*Hes1*], [*Fhl1*], [*Hmgn2*] modules relative to $C_1$. (**I**) Cells in different states are predicted to respond differently to morphogens. Plot showing the percentage of models (y-axis) where states $C_1$ and $C_3$ (x-axis) transition to $C_6$ (characterized by unique marker gene module [*Msx2*]), in response to [LIF]+[BMP]. $C_1$ cells remain stable in response to [LIF]+[BMP] signaling in >98% of the models whereas $C_3$ cells are destabilized and move to the $C_6$ state in ~11% of the models.

The following source data and figure supplements are available for figure 4:

**Source data 1.** Gene modules used for modeling the network.
**Source data 2.** Binary expression profiles of the gene modules used for modeling the network in the 9 cell clusters.
**Figure supplement 1.** Summary of gene modules and illustration of production rate determination for each gene module.
**Figure supplement 2.** Summary of parameters for model gene regulatory network.
**Figure supplement 3.** The predictions of the gene regulatory network are robust to changes in the probability threshold for considering a gene to be a transition or a marker gene.

On the other hand, [*Snai1*] and [*Oct4*] are not expressed together in any of the nine cell states. We investigated the predicted effects of [*Snai1*] overexpression on [*Oct4*] in the epiblast-like state $C_1$ and mesendoderm-like state $C_2$, both of which normally express [*Oct4*] but not [*Snai1*]. Although [*Snai1*] has a negative interaction with [*Oct4*] in 79.2% of the models, the modules expressed in $C_1$ exert a greater positive drive on [*Oct4*] (*Figure 4E and F*) than those expressed in $C_2$. This leads to the prediction that [*Oct4*] is less sensitive to [*Snai1*] overexpression in state $C_1$ compared to $C_2$ (*Figure 4G*).

We next considered the effect of morphogen signals in different states. Specifically, we considered the LIF, BMP, WNT and FGF signaling pathways, which are known to play a significant role in patterning the early embryo, as well as are central to our in vitro differentiation process (Materials and methods). We grouped signaling genes by their respective pathways (defined by GO categories) and assigned each group to a module based on its average expression pattern across the nine cell states. Because WNT and FGF modules show no changes in expression across all cell states (most likely due to the large number of genes that fall into the relevant GO categories), we focused on investigating the effects of LIF and BMP signaling on cells in the epiblast-like $C_1$ and in the bi-potent ectoderm-like state $C_3$ (*Figure 4H*). Given an initial state $C_1$ or $C_3$, we calculated the probabilities that cells either remain in the same state or move to a different state in response to [LIF] and [BMP] (Materials and methods). Our simulations found that for ~98% of the models, cells that are initially in state $C_1$ either remained stabilized in $C_1$ or moved to state $C_0$ in response to [LIF] and [BMP] addition. However, in response to the same perturbation, the vast majority of cells in the $C_3$ state either transitioned to the neural crest-like state $C_6$(11.2%) or stayed in the C3 state (86.1%) (*Figure 4I*).

To summarize, we predict that [*Oct4*] expression is less sensitive to [*Sox2*] overexpression in state $C_0$ than in $C_1$; [*Oct4*] expression is less sensitive to [*Snai1*] overexpression in state $C_1$ compared to $C_2$; and cells in state $C_3$, but not in $C_1$, can transition to state $C_6$ following [LIF]+[BMP] exposure.

Importantly, we further noted that the model predictions were robust to changes in the probability cutoff for the genes we considered: although the number of gene modules changed (27 modules for a cut off of 0.7 and 24 for 0.9), we found that the models made the same qualitative predictions (*Figure 4—figure supplement 3*).

Thus, by categorizing genes into different modules by their expression patterns across the observed cell states, these modules provide a starting point for modeling the gene regulatory network responsible for cell fate decisions, allowing us to make predictions for how the network gives rise to distinct phenotypic responses to the same perturbation across different cell states.

## Interpretation of *Sox2*, *Snai1*, and LIF+BMP are cell state dependent

We next experimentally tested the qualitative aspects of the model's predictions of state-dependence in cells' responses to perturbations. We first tested how cells' Oct4 levels respond to *Sox2* overexpression in the naïve ES and epiblast-like states $C_0$ and $C_1$. We transiently transfected cells with a plasmid containing a Tet-inducible bi-directional promoter, flanked by the open reading frames of *Sox2* and mCerulean, which we used as a fluorescent reporter of induction (*Figure 5—figure supplement 1A*). We induced overexpression in cells either in the undifferentiated $C_0$ state or the epiblast-like $C_1$ state, which correspond to Day 0 and Day 2 of differentiation, respectively (*Figure 3D*, *Figure 5—figure supplement 1D*). As a control, we used identical populations that were transfected with a plasmid containing only mCerulean under the inducible promoter. In such experiments, we typically saw mCerulean fluorescence appear approximately three hours into induction and persist for about three to four days after transfection. We therefore induced overexpression for 24 hr to minimize the effect of plasmid loss but still allow for several cell cycles to occur during induction. Following induction, we fixed and immunostained the cells for Oct4, and analyzed the results via flow cytometry. In agreement with our predictions (*Figure 4D*), we found that *Sox2* overexpression correlates ($R = -0.3258$, $p = 1.48 \times 10^{-13}$) with downregulation of Oct4 in the epiblast-like state $C_1$ (significant relative to control, $p = 5.72 \times 10^{-31}$; see also *Figure 5—figure supplement 1C*), whereas this effect was not observed in undifferentiated cells (state $C_0$) (*Figure 5A and B*).

We then tested the effects of *Snai1* overexpression on Oct4 in the epiblast-like state $C_1$ and mesendoderm-like state $C_2$, using the same experimental framework as described above. On day three of differentiation, cell populations either contain a mixture of $C_1$, $C_2$ and (minimally) $C_4$ cell states, or a combination of $C_1$, $C_3$ and $C_5$ (or $C_6$), depending on the signaling conditions (*Figure 3D*). Using the signaling conditions that yield the former set of cell states ($C_1$, $C_2$ and $C_{4)}$, we transfected cells at 2.5 days into differentiation, and drove overexpression of *Snai1* 12 hr later in a population consisting primarily of cells in $C_1$ and $C_2$ states (*Figure 5—figure supplement 1D*). After 24 hr of *Snai1* overexpression and further differentiation, we fixed and immunostained the cells for T to distinguish cells in $C_1$ (T-low) and $C_2$ (T-high) states. We also immunostained the cells for Oct4 to distinguish the $C_1$ state from other T-low states that arise during the last 24 hr of differentiation following the initiation of induction. We found that the fraction of $C_1$ cells within the transfected population was significantly reduced relative to control ($p = 1.98 \times 10^{-13}$), suggesting that cells in this state had downregulated Oct4 levels in response to *Snai1* overexpression. On the other hand, the fraction of $C_2$ cells within the transfected population and their Oct4 levels were maintained relative to control, in agreement with our predictions (*Figure 4G*; *Figure 5C and D*).

Finally, we tested whether cells in epiblast-like $C_1$ and bi-potent ectoderm-like $C_3$ states respond differently to LIF+BMP signaling, as predicted by our model. In order to investigate the relationship between a cell's initial state and its final state in response to LIF+BMP exposure, we needed to assess cells' initial states non-invasively. We found that 2.5 days into differentiation, we could obtain populations that consist primarily of cells in epiblast-like state $C_1$ and bi-potent ectoderm-like state $C_3$ (*Figure 5—figure supplement 1E*), which have high and low expression of Oct4, respectively. We therefore utilized an Oct4-mCitrine mES cell line that we had previously engineered (*Thomson et al., 2011*) to distinguish cells in $C_1$ and $C_3$ states after 2.5 days of differentiation. At this point, 1200 U/mL LIF and 25 ng/mL BMP4 were added to the media, after which we followed individual cells' Oct4 expression dynamics for approximately 24 hr via live-cell microscopy, followed by fixing and immunostaining for Msx2, a unique marker gene for the neural crest-like cell state $C_6$ (*Figure 5E and F*). As predicted by the model (*Figure 4I*), only cells that had low Oct4 levels (and were therefore in the bi-potent ectoderm-like state $C_3$) prior to LIF+BMP exposure showed

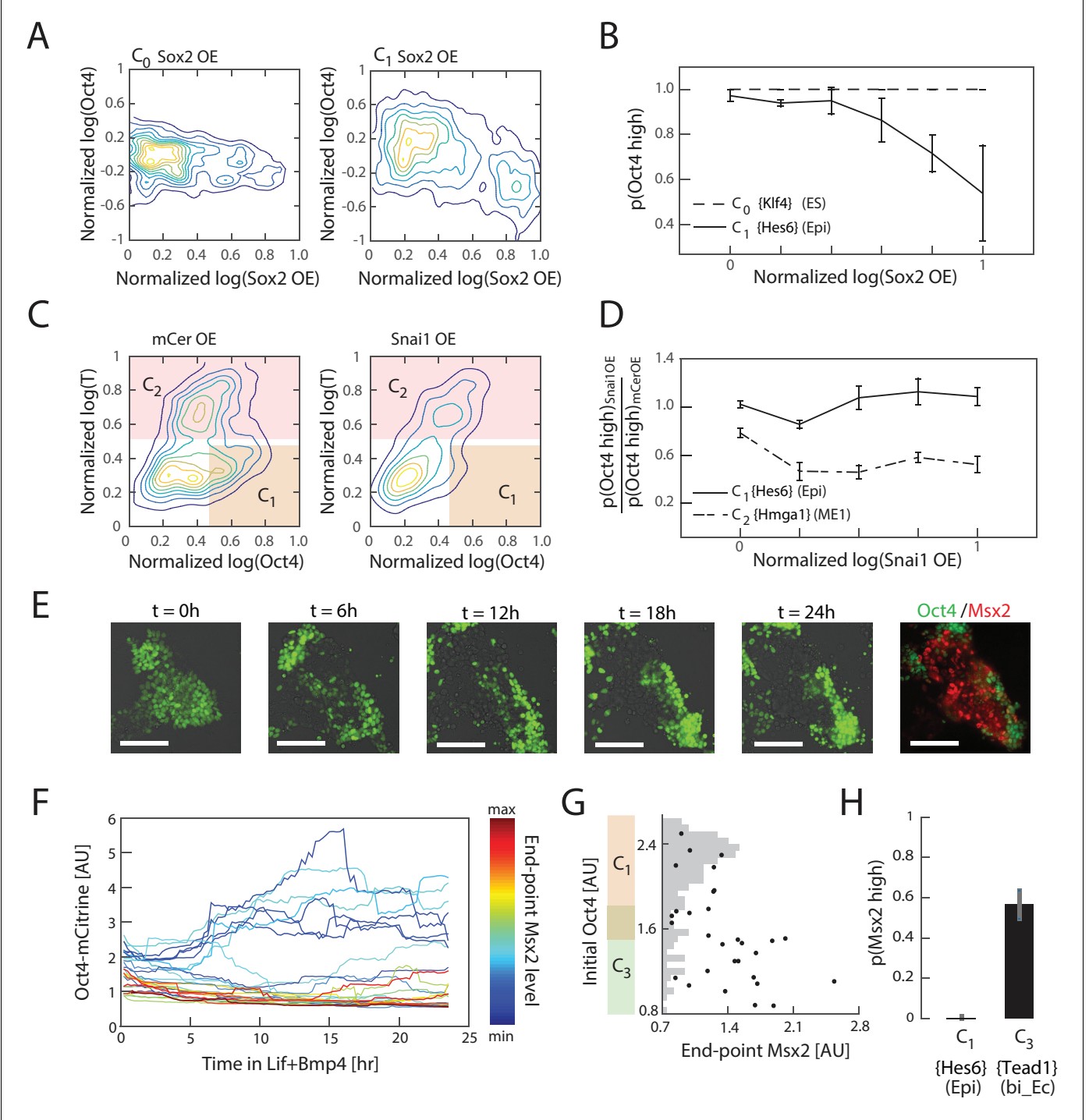

**Figure 5.** Experimental validation shows that interpretation of *Sox2*, *Snai1*, and LIF+BMP is cell state dependent. (**A**) Comparison of the effects of *Sox2* overexpression (x-axis) on Oct4 levels (y-axis) in the naïve ES state $C_0$ (left) and epiblast-like $C_1$ state shows negative correlation between *Sox2* overexpression and Oct4 levels in the $C_1$ state, but not in $C_0$. Plots showing mCerulean (marker) -only overexpression in $C_0$ or $C_1$ are indistinguishable from *Sox2* overexpression in $C_0$ (*Figure 5—figure supplement 1C*). (**B**) Fraction of Oct4-high cells (y-axis; defined as greater than $2\sigma$ below the mean log of Oct4 of non-transfected control cells) plotted against binned *Sox2* overexpression level confirms model prediction (*Figure 4D*) that *Sox2* overexpression leads to downregulation of Oct4 in $C_1$ but not $C_0$. (**C**) Comparison of the effects of *Snai1* and mCerulean-only (left) overexpression on Oct4 levels (x-axis) in the epiblast-like $C_1$ and mesendoderm-like $C_2$ states (y-axis; T-low and –high, respectively) shows downregulation of Oct4 in response to *Snai1* overexpression in the $C_1$ state but not in $C_2$. (**D**) Fraction of Oct4-high cells in *Snai1* overexpressing cells, normalized by this fraction in mCerulean overexpressing control cells (y-axis), plotted against binned *Snai1* overexpression level (x-axis) confirms the prediction (*Figure 4G*) that

*Figure 5 continued on next page*

*Figure 5 continued*

*Snai1* overexpression leads to greater downregulation of Oct4 in $C_2$ compared to $C_1$. (E) Live cell images of Oct4-mCitrine cells at t = 0, 6, 12, 18, 24 hr of LIF+BMP exposure. At t = 0, cells are either in state $C_1$ (Oct4-high) or $C_3$ (Oct4-low) (*Figure 5—figure supplement 1E*). (Scale bar = 100 µm) Cells were fixed at t = 24 hr and immunostained for *Msx2*. (F) Time series (x-axis) traces of single-cell Oct4 expression (y-axis) taken every 15 min from live cells. Each trace is colored by its relative end-point *Msx2* immunofluorescence intensity level. (G) The initial Oct4 reporter (mCitrine) intensity (y-axis) and final Msx2 immunofluorescence (x-axis) are negatively correlated. Each dot represents a single cell. Histogram of Oct4 reporter intensity at t = 0 levels shown in gray. Based on this histogram, we defined a range of threshold values for determining Oct4-high and –low (shown in overlapping region of orange and green along y-axis). (H) Plot showing fraction of Msx2-high (y-axis; as defined by greater than 2σ above background) confirms prediction (*Figure 4I*) that Msx2 is upregulated with a greater probability in the $C_3$ state compared to $C_1$ (x-axis) in response to LIF+BMP exposure.

The following figure supplement is available for figure 5:

**Figure supplement 1.** Controls for perturbation experiments.

upregulation of Msx2 in response to LIF+BMP ($R = -0.5056$, $p = 0.0044$, *Figure 5G and H*). Together, these results show that the inferred cell states reflect phenotypic discreteness in cells' responses to perturbations, and that the gene expression changes that define these responses mirror those predicted by our model gene regulatory network.

## Discussion

By using learned sparse patterns of gene expression from established experimental systems (*Furchtgott et al., 2016*), we can analyze single-cell transcriptomics data to uncover the gene expression dynamics of differentiation. This method naturally identifies a small set of transcription factors whose expression profiles are multimodal across neighboring cell states. Given that transcription factors are key orchestrators of gene expression and therefore cell fate decisions (*Spitz and Furlong, 2012*), multimodal distributions of the expression levels of even a small set of transcription factors can define cell states in a population of cells.

While cell states can be characterized by the gene expression patterns of key sets of genes, these states can only be fully validated by demonstrating distinct physiological properties. To discover distinct properties of the cell states in early mES cell differentiation, we built probabilistic models of the underlying network. Requiring these models to have discrete cell states leads to the prediction that each cell state has a distinct response to perturbations by signals and changing levels of gene expression. Thus, the cell states we discovered can be functionally defined by their responses to perturbation. Our experimental tests show, as predicted by the model network, that *Oct4* is either downregulated or unaffected by overexpression of *Sox2* or *Snai1*, depending on the cell state. Previous studies have already shown that *Sox2* and *Oct4*, along with *Klf4*, constitute part of a positive feedback loop that stabilizes the pluripotent ground state (*Kim et al., 2008*; *Young, 2011*). It is also known that in undifferentiated cells, *Snai1* overexpression leads to downregulation of *Oct4* expression and, subsequently, to exit of pluripotency (*Galvagni et al., 2015*). However, our results demonstrate that these interactions are state-dependent by showing that the effective positive interactions between *Sox2* and *Oct4* become destabilized as *Klf4* levels drop and cells transition to a primed, epiblast-like pluripotent state. Similarly, the negative interaction exerted by *Snai1* on *Oct4* becomes attenuated in the presence of early primitive streak genes such as *T*. We also predict and show that LIF+BMP exposure pushes bi-potent ectoderm-like cells toward an *Msx2*-positive neural crest-like state, but this effect is not seen in epiblast-like cells. These results are further supported by the fact that both LIF and BMP signaling pathways can be used to keep cells in the pluripotent cell state (*Chambers, 2004*; *Tam et al., 2006*; *Ying and Smith, 2003*), and that BMP signaling plays a significant role in the differentiation of neural crest cells (*Knecht and Bronner-Fraser, 2002*). Together, these findings signify that the inferred cell states directly reflect differences in cells' responses to perturbations and show that these cell states can also be defined by their unique responses to perturbations.

Comprehensive interrogation of gene expression through RNA sequencing is impossible without the termination of cells, providing only static snapshots of gene expression during differentiation. Despite this and the complexity of the underlying network, we discover that both cell states and the

sequence of cell state transitions can be accurately determined by monitoring the levels of just a few transition or marker genes. Monitoring the expression dynamics of these key genes in live cells using microscopy will allow us in the future to continuously track the cell-fate decisions of individual cells. The inferred gene modules therefore represent the 'order parameters' by which cell-state transition dynamics can be directly measured. Live cell microscopy experiments will also allow us to measure, in conjunction with cell state transition dynamics, changes in individual cells' spatial environment, movement, lineage history, and cell cycle dynamics in order to address fundamental biological questions as to how these factors affect cell fate decisions. Finally, our results suggest that cell-to-cell heterogeneity within differentiating populations arises largely as a consequence of cells' variability in their timing of cell state transitions. Our inferred cell clusters show mixing of cells from different time points (*Figure 1—source data 1*, *Figure 2—source data 1*), suggesting that the observed states themselves do not change over time and that at the population level, differentiation occurs as a change in the proportions of cells in various cell states rather than through changes in the cell states themselves (*Figure 3D*). Since cells interpret perturbations differently even in consecutive states (*Figure 5*), this suggests that heterogeneity arising from timing variability is further amplified in response to signal addition or fluctuations in gene expression level. These findings emphasize the importance of understanding how the timing of cell state transitions is controlled during development.

## Materials and methods

### Clustering and re-clustering using seurat

Clustering was performed using Seurat (*Satija et al., 2015*). For the initial seed clustering, we applied Seurat to the gene expression of all 2672 transcription factors for the 288 single cells. For subsequent re-clustering steps, clustering was performed on a reduced set of genes for which $p\left(\alpha_i = 1 \text{ or } \beta_i = 1 \middle| \left\{g_i^{A,B,C}\right\}, T, \{C\}\right) > 0.5$ for at least one triplet at the previous iteration (assuming a prior odds of $O_{\beta|T}(i) = 5 \times 10^{-2}$). This reduced set contained between 800 and 1050 genes at each of the reclustering steps (*Figure 2—figure supplement 2A*).

Seurat performs spectral t-SNE on the statistically significant principal components (PCs) of the gene expression dataset, and it determines the significance of each PC score using a randomization approach developed by Chung and Storey (*Chung and Storey, 2015*). Our initial seed clustering was performed using the first 10 PCs; subsequent re-clusterings used the first 8 PCs.

Finally, Seurat performs density-based clustering on the t-SNE map; we used a density parameter of G = 8 (*Macosko et al., 2015*).

### Convergence of clustering configurations from different seed configurations

In order to test that our results were robust to the choice of seed clusters, we further used k-means clustering, a standard clustering method, which has previously been applied to identify different cell types using single-cell transcriptomics data (*Buettner et al., 2015*).

We start with a seed clustering configuration of 12 clusters $\left\{c_1^0, c_2^0, \ldots, c_{12}^0\right\}$ obtained using k-means clustering, which is distinct from the seed clustering configuration obtained via Seurat (*Satija et al., 2015*). The number of clusters was determined using the gap statistic (*Tibshirani et al., 2001*). We obtained 164 sets of transitions between clusters and identified 981 transcription factors that were high probability (probability >0.5) marker or transition genes for at least one of the identified transitions. We next re-clustered the single cells in the gene expression space defined by these 981 marker or transition genes, using k-means clustering, to obtain a new cluster set $\{C_1\} = \left\{c_1^0, c_2^0, \ldots, c_{10}^0\right\}$, consisting of 10 clusters. In the next iteration, the number of clusters went down to 9, and so on. By iteratively determining the most likely sets of transitions, the corresponding most likely marker and transition genes and re-clustering the cells within the subspace of these genes, our algorithm converged upon the most likely set of cell clusters (*Figure 2A*). We found that the eventual clustering configurations obtained using k-means clustering and Seurat are the same, confirming that the seed clusters do not affect the final outcome (*Figure 2—figure supplement 2A, B and C*).

## Framework for quantitative modeling of germ layer differentiation

Classifying genes based on their patterns of expression along the inferred lineage tree rather than by gene-gene correlations allowed us to identify gene modules (which included the transition and marker genes we inferred as well as signaling genes: BMP, WNT, LIF, see Tables S4 and S5) with similar expression patterns in successive cell-fate decisions.

### Determination of gene modules

We obtained 321 transcription factors from the triplets along the tree and classify them based on their pattern across the triplets. In order to explain the discretization procedure let consider the example of Otx2, which is a transition gene for the triplet involving $C_1$, $C_2$ and $C_3$ clusters, where $C_1$ is the intermediate cluster. Since Otx2 is expressed at high levels in cluster $C_1$ and $C_3$ and is downregulated in cluster $C_2$, we assigned it a value of 1 in clusters $C_1$ and $C_3$ respectively and 0 in cluster $C_2$. We then repeated this local binarization process across all triplets along the lineage tree. We grouped all the genes that showed the same locally binarized expression pattern as Otx2 and obtained their average expression level across all the other clusters. Subsequently, we assigned these genes a value of 1 in a cluster if the average expression of these genes in that cluster was comparable (within ~10% of the mean) or higher than the lower value of their average expression level in the $C_1$ and $C_3$ clusters. Some genes, such as Oct4 and Etv5 are re-used at multiple branching points i.e. they belong to multiple triplets, either as marker genes or transition genes, and hence belong to different groups (*Figure 4—figure supplement 1*). Certain genes that are re-used exhibit three distinct levels of expression. For instance, Sox2 comes up as a marker gene for $C_0$ cluster, when we consider the triplets involving clusters $C_0$, $C_1$ and $C_2$ and $C_0$, $C_1$ and $C_3$ clusters respectively. However, it also acts as a transition gene for the triplet involving $C_1$, $C_2$ and $C_3$ clusters, where Sox2 is downregulated in $C_2$. Such a gene expression pattern would require three distinct levels (high in $C_0$, medium in $C_1$, $C_3$, and low in $C_2$). We classified the medium and higher expression level as one and low expression level as 0. It must be noted that we determined binary gene expression profiles by calculating the mean log2 fold-change in expression level for each group of genes. This way we acquired a total of 29 modules with unique binary gene expression profiles. We denote each module by a representative gene; the genes that belong to each module are shown in *Figure 4—source data 1*.

### Local-field gene regulatory network model for gene modules

In order to build a quantitative model relating the gene modules, we write a N-component gene regulatory network governed by a set of differential equations:

$$\dot{m}_i = -\frac{m_i}{\tau_i} + r_i^0 + r_i\left(\vec{m}\right) \qquad (i = 1, \ldots, N) \tag{1}$$

where $\tau_i$ and $r_i^0$ are respectively the life-time and basal production rate of module $i$; we will rescale $\tau_i = 1$ and $r_i^0 = 0$ without any loss of generality. We denote the level of module $i$ as $m_i$. We assume here that modules interact only by modulating each-other's rate of production, described here by rate functions $r_i\left(\vec{m}\right)$ which depend on the state $\vec{m} = [m_1, \ldots, m_N]$ of the gene regulatory network.

As above, we consider that the production rate $r_i\left(\vec{m}\right)$ is the result of only direct interactions, in which each gene $j$ exerts a drive on gene $i$ which is equal to an interaction strength $J_{ij}$ (positive or negative) multiplied by the level of module $j$. The total drive $\phi_i$ on gene $i$ is the sum of the drives from the different modules:

$$\phi_i\left(\vec{m}\right) = \sum_{j=1}^{N} J_{ij} m_j \tag{2}$$

We now assume $r_i$ has a universal scaling form that is the same for all factors,

$$r_i\left(\vec{m}\right) = r[\mu(\phi_i - \phi_0)] \tag{3}$$

where $r(\phi; \phi_0, \mu)$ is a monotonic sigmoidal function centered at $\phi_0$ and bounded by the limits

$$r(\phi) = \begin{cases} 0, & \phi \ll \phi_0 \\ 1, & \phi \gg \phi_0 \end{cases} \tag{4}$$

the sharpness of crossover is determined by the nonlinearity parameter $\mu$. The upper bound of $r_i = 1$ sets the maximum sustainable expression at $m_i = 1$. In the limit $\mu \to \infty$, $r(\phi)$ becomes the Heaviside step function, and $m_i \in \{0, 1\}$ is binary.

Suppose state $\vec{m}^{\alpha} = \{m_1^{\alpha}, \ldots, m_{29}^{\alpha}\}$ with expression level $m_i^{\alpha}$ in module $i$ is a stable state of the network. In the limit $\mu \to \infty$, the condition for $\vec{m}^{\alpha}$ to be a fixed point is:

$$m_i^{\alpha} = H\left(\sum_j J_{ij} m_j^{\alpha} - \phi_0\right) \quad m_i^{\alpha}, m_j^{\alpha} \in \{0, 1\} \tag{5}$$

where $H$ is the Heaviside step function. (Note that if $\phi_0 > 0$ then $\vec{m} = \vec{0}$ is always a stable fixed point of the network.)

In this limit, each state $\vec{m}^{\alpha}$ of the network is associated with N constraints given by inequalities of the form

$$m_i^{\alpha} = 0 \;\Rightarrow\; \sum_j J_{ij} m_j^{\alpha} < \phi_0 \tag{6}$$

$$m_i^{\alpha} = 1 \;\Rightarrow\; \sum_j J_{ij} m_j^{\alpha} > \phi_0 \tag{7}$$

If $\vec{m}^{\alpha}$ is a fixed point, all N of its constraints must hold. If we know the fixed points of the network, then we can write down a system of inequalities that constrain possible values for $J_{ij}$. Since gene-gene interactions cannot be infinitely strong, $J_{ij}$ must be bounded. We take $|J_{ij}| < 1$ and $\phi_0 = 0.1$. We further vary the value of the critical drive $\phi_0$ from $-2$ to $2$ to check the robustness of the predictions. We find that all the results qualitatively hold although the individual probabilities change.

## Linear programming

The constraints (7) and (8) placed on $J_{ij}$ by the fixed point condition are linear in $J_{ij}$. We can take advantage of this fact and use linear programming methods (**Gass, 2013**) to obtain solutions for $J_{ij}$ by extremizing a linear objective function of the form

$$U(J_{ij}) = \sum_{i,j} a_{ij} J_{ij} = \text{constant} \tag{8}$$

where $a_{ij}$ are constant coefficients. The system of constraints defines a $N^2$-dimensional polytope in $J$-space that encloses all solutions of $J_{ij}$ consistent with the fixed-point constraints, and $U$ defines a $N^2 - 1$ dimensional hyperplane. Linear programming returns a solution for $J_{ij}$ (a point in $J$-space) where the polytope contacts a $U$-plane of extremal value. The solution will lie on the boundary of the polytope and is in general non-unique. There is no general principle with which to select any specific $U$-plane as the 'best' objective function. Furthermore, one would like to sample points in the interior of the polytope, and not just on its surface. Here, guided by the fact that we seek pertubative solutions for $J_{ij}$ that ideally lie close to the origin, we impose a fictitious additional constraint on the polytope in the form of a hyperplane that contains the origin

$$\sum_{i,j} a_{ij} J_{ij} \leq 0, \qquad a_{ij} \in \{0, 1\} \tag{9}$$

where the coefficients $a_{ij}$ are randomly chosen; this in effect slices the polytope in two and exposes an interior plane. Then, using the same choices of $a_{ij}$ to define a $U$-plane, we seek a linear programming solution that maximizes $U$, that is, a solution that lies on the now-exposed interior plane (if possible). Because these fictitious constraints radiate from the origin, points in the polytope that lie closest to the origin are sampled more densely.

## Common features of the sampled networks

By using many different randomly generated fictitious constraints to sample the polytope, we can study the ensemble of model networks that all satisfy the fixed point constraints (*Figure 4—source data 2*), and attempt to determine whether they share any common regulatory motifs. As discussed in the main text, we sampled 10,000 solutions $J_{ij}$ that satisfied the fixed-point constraints defined by the binarized expression patterns of the known cell states. We then calculated the mean and coefficient of variation (c.v.) for each coupling. We were thus able to discover a core network between the different modules that is shared by the majority of solutions (*Figure 4A*).

## Predictions for Sox2 and Snai1 overexpression

Our model makes predictions for what happens to the level of Oct4 when Sox2 and Snai1 are over-expressed in different cell states. Sox2 and Oct4 are both present in the $C_0$ and $C_1$ clusters. On the other hand, Snai1 is not present in $C_1$ and $C_2$ but Oct4 is present in both clusters. We perturb the Sox2 and Snai1 levels by amounts $\Delta s$ in the above mentioned states, which lead to a change in the field $\phi_i$ total drive on Oct4 level. Numerically we vary $\Delta s$ in steps of 0.1 and for each step compute the number of models out of the 10000 total models, for which the Oct4 level decreases to zero. From this number we obtain the fraction of models for which the level of Oct4 goes down.

## Predictions for BMP and LIF addition

In order to predict the effect of morphogen signals in different cell states, we considered the LIF, BMP, WNT, and FGF signaling pathways, which are known to play a significant role in patterning the early embryo. We assumed that no single gene in each given pathway is sufficient to evoke a signaling response, but a response rather requires the combined presence of the various constituent genes of the pathway. We therefore grouped genes by their respective signaling pathways and assigned each group to a module based on its average expression pattern across the nine cell states. The discretization process of this mean expression pattern was the same as that used for TF genes. The signaling genes we used are shown in *Figure 4—source data 1*.

We next modeled the dynamics of BMP and LIF addition. By construction, the nine observed cell states (and the null state $\vec{m} = \vec{0}$) are fixed points for all 10,000 sampled solutions for $J_{ij}$. However, each solution $J_{ij}$ may have additional spurious fixed points. However, given that we only see 9 cell states, we would expect the spurious states to be unstable. In order to overcome this problem, we used the following method.

Given a particular solution $J_{ij}$, any arbitrary state of the network $\vec{m}$ (not necessarily a fixed point) will have dynamics obeying

$$m_i(t+1) = H\left(\sum_j J_{ij} m_j(t) - \phi_0\right) \tag{10}$$

where $m_i(t)$ and $m_i(t+1)$ are the levels of module $i$ at successive discretized time points.

For each particular solution $J_{ij}$, cells will get stuck in spurious fixed points; yet these spurious fixed points are highly unlikely to exist since they are stable in only a small number of the sampled $J_{ij}$. We can capture the average dynamics of different states of the network given the set of sampled solutions $\{J_{ij}\}$ by calculating the probability over all sampled solutions of moving from one arbitrary state $\vec{m}^a$ to another arbitrary state $\vec{m}^b$. This allows us to define a $2^{29} \times 2^{29}$ state-to-state transition matrix $T$:

$$T_{b \leftarrow a} = p\left(\vec{m}^a \rightarrow \vec{m}^b | \{J_{ij}\}\right) \tag{11}$$

If we denote as $\vec{p}(t)$ the vector of probabilities of being in the $2^{29}$ different states at time $t$, then

$$\vec{p}(t+1) = T\,\vec{p}(t) \tag{12}$$

In order to figure out what happens to cells in different states to BMP and LIF addition, we calculated the probability of moving between fixed points $\vec{m}^\alpha$ and $\vec{m}^\beta$ when overexpressing some set of

modules $\{m_i\}$. We calculated the dynamics using the transition matrix $T$ and enforced the overexpression of the set of modules (BMP and LIF module respectively) at each time point, updating the probabilities $\vec{p}(t)$ accordingly. The probabilities shown in *Figure 4* are after 1000 time steps.

## ES-cell culture

v6.5 (RRID: CVCL_C865; passage number 18 ~ 30; mycoplasma tested negative) mouse embryonic cells were maintained and passaged in monolayer (non-embryoid body formation) in N2B27 basal media with signaling molecules and/or small molecules added to the basal media. ES cells were maintained in a pluripotent cell state using 1200 U/mL mLIF (murine leukemia inhibitory factor), 1 µM PD0325901 (MEK inhibitor), and 3 µM CHIR99021 (GSK inhibitor) conditions (a.k.a. 'LIF + 2i'; *Ying et al., 2008*), and passaged every two days. To passage cells, we added 0.01% trypsin to cells after aspirating media and incubated the plate in 37'C for 1 ~ 2 min to detach cells. The trypsin was then quenched with 0.5 mL of fetal bovine serum, and the resulting cell suspension was collected, counted, and pelleted at 200 x g for 5 min at room temperature. The supernatant was aspirated and the cells were resuspended and re-seeded onto a gelatinized tissue culture dish at a density of 1e6 cells per 10 cm diameter plate. All cell lines were depleted of feeders and transitioned to serum free medium over several passages prior to experiments (*Ying and Smith, 2003*). N2B27 is prepared as described in *Gaspard et al. (2008)*, *Ying and Smith (2003)*.

## ES cell differentiation

Cells were seeded at a density of $10^6$ per 10 cm diameter plate, and were not trypsinized again until they were harvested for analysis. We either exposed cells to 0.4 µM PD0325901 or 3 µM CHIR99021 and 10 ng/mL Activin A (human, rat, mouse) for 2 days or 3 days, respectively, followed by either 25 ng/mL *hBmp4* or 1 µM LDN193189 (BMP antagonist) for up to two days. Media was replenished every 48 hr. Cells exposed to 0.4 µM PD0325901 gave rise to ectodermal lineages, as characterized by expression of *Sox1*, *Pax6* (treated with LDN193189), *Slug*, and *Msx2* (treated with *hBmp4*) after three days of differentiation. Cells exposed to CHIR99021 and Activin A gave rise to mesendodermal lineages (*Sumi et al., 2008*), as characterized by expression of *T* after three days of differentiation, and *FoxA2* (treated with LDN193189) and *Gata4* (treated with *hBmp4*) after four days of differentiation.

## Single-cell RNA-Seq

CEL-seq libraries as previously reported (*Hashimshony et al., 2012*) with a few modifications. Single cells were sorted with a FACSAria into 96 well plates containing 1.2 µL 2 × CellsDirect Buffer (Life Technologies) with 0.1 µL of ERCCs diluted to $1 \times 10^{-6}$ molecules (Life Technologies). Plates were frozen and stored at $-80°C$. For library preparation, mRNA was reverse transcribed using 0.15625 pmol of oligoT primer carrying a cell-specific 8 NT barcode and a 5 NT unique molecular identifier (UMI) (*Islam et al., 2014*). Barcode design ensured at least two nucleotide differences from any other barcode. Samples were lysed at 70°C for 5 min, then reverse transcribed using Superscript III for two hours at 50°C, then primers digested with 1 µL of ExoSAP-IT (Affymetrix). Second strand synthesis was carried out with Second Strand Synthesis Buffer, dNTPs, DNA Polymerase, and RNAse H (NEB) at 16°C for 2 hr. Single-cell cDNAs were pooled by 24 wells per library, with each library containing a water-only well and one ERCC-only well. Pools were purified with an equal volume of RNA Clean Beads (Beckman Coulter) and amplified at 37°C for 15 hr using the HiScribe T7 High Yield RNA Synthesis kit (NEB), and treated with DNAse I (Life Technologies). Amplified RNA was fragmented using the NEBNext RNA Fragmentation Module (NEB), purified with an equal volume of RNA Clean Beads, and visualized using the RNA Pico Kit on the Bioanalyzer 2100 (Agilent). The RNA fragments were repaired with Antarctic Phosphatase and Polynucleotide Kinase (NEB), and purified using an equal volume of RNA Clean Beads. cDNA libraries were made using the NEBNext Small Library Prep Kit according to the manufacturer's instructions, except Superscript III was used for the RT step. Index primers were used in PCR amplification. Approximately 160–200 nmol of a pool of libraries were size selected to exclude species smaller than 180 bp on a 2% Dye Free cassette on the Pippin Prep (*Roccio et al., 2013*) and concentrated to approximately 14 µL. Pools were then quantified by qRT-PCR using p5 (5'-AATGATACGGCGACCACCGAGA-3') and p7 (5'-CAAGCAGAA-GACGGCATACGAGAT-3') primers and by Bioanalyzer (DNA High Sensitivity Kit, Agilent), and

sequenced on an Illumina HiSeq. The custom sequencing primer: 5'-TCTACACGTTCAGAGTTC TACAGTCCGACGATC-3' was included with Illumina primer HP10 for sequencing. Standard Illumina primers HP12 and HP11 were used for the index read and the transcript read, respectively. PE50 kits (Illumina) were used for sequencing with read lengths of 25 nt, six nt, and 47 nt for read1 (cell barcode, UMI), index (library), and read2 (transcript), respectively. Following quantification, we discarded the data from wells that yielded below a total of 20,000 UMI (threshold based on empty well controls), which left us with 358 cells. Further, as others have recognized (*Paul et al., 2015*), we found that some well-to-well mixing was present with CEL-Seq multiplexed single-cell RNA-Seq. We used the data only from 288 cells because of this mixing artifact. The raw and processed RNA-seq data for the 288 cells can be found on the GEO database (accession number: GSE105054).

## Immunofluorescence

Cells were grown on ibidi μ-bottom plates and fixed with 4% paraformaldehyde. Cells were permeabilized with ice-cold 100% methanol, blocked with 5% donkey serum, incubated with primary antibody, washed, and incubated with DAPI and secondary antibody coupled to Alexa488 Alexa568, or Alexa647. Images were acquired with a Zeiss 40× plan apo objective (NA 1.3) with the appropriate filter sets. Data was analyzed using custom written code in MATLAB. Antibodies and dilutions used in this study: *Klf4* (Abcam ab129473, 1:400); *Nanog* (eBiosciences 14–5761, 1:800); *Oct4* (Santa Cruz sc-8628, 1:800; Cell Signaling 2840, 1:400); *Sox2* (eBiosciences 14–9811, 1:800); *Otx2* (Neuromics GT15095, 1:400); *T* (*Brachyury*) (Santa Cruz sc-17745, 1:200); *FoxA2* (Cell Signaling 8186, 1:400); *Gata4* (eBiosciences 14–9980, 1:400); *Sox1* (Cell Signaling 4194, 1:200); *Pax6* (DSHB Pax6, 1:200); *Msx1 +2* (DSHB 4G1, 1:200); *Slug* (Cell Signaling 9585, 1:200), *Snai1* (Cell Signaling 2879, 1:200).

## Live-cell microscopy

For live-cell time-lapse microscopy, cells were plated into N2B27 without phenol-red (plus signaling molecules and small molecules) on ibidi μ-bottom plates. Cells were imaged on a Zeiss Axiovision inverted microscope with a Zeiss 40× plan apo objective (NA 1.3) with the appropriate filter sets with an Orca-Flash 4.0 camera (Hamamatsu). The microscope was enclosed with an environmental chamber in which $CO_2$ and temperature were regulated at 5% and 37°C, respectively. Images were acquired every 15 min for 12–48 hr. Image acquisition was controlled by Zen (Zeiss); image analysis was done with ImageJ (NIH) and Matlab (MathWorks). 38 HE GFP/43 HE DsRed/46 HE YFP/47 HE CFP/49 DAPI/50 Cy5 filter sets from Zeiss. Transition duration of Otx2-mCitrine cells was defined as the time between the last image at which a cell's reporter intensity was equal to or below its intensity at t = 1 and the first image at which its intensity was equal to or above 2.2 (mean – $\sigma$ of upper mode of Otx2 reporter intensity) on the normalized scale.

## Plasmid transfection

We cloned *Sox2* or *Snai1* cDNA to one side of a bi-directional Tet-on promoter (pTRE3G-BI; Clontech), to the other side of which we had cloned in mCerulean cDNA. Mini-prepped plasmid was ethanol-precipitated to further concentrate and remove any possible endotoxins. For *Sox2* overexpression, cells were seeded at 100,000 cells per 35 mm diameter plate in 2 mL of either LIF +2i conditions or differentiation media (0.4 μM PD0325901 or 3 μM CHIR99021) for 1 day. 200 μL of FBS was then added to each plate and 1.8 ug of plasmid was transfected using 5.4 μL of JetPrime (Polyplus). Cells were incubated for 12 hr, then washed with PBS and replenished with fresh LIF+2i or differentiation media. We then added 3 μL of Tet-Express mixed with 2.5 μL of Intensifier reagent (Clontech). Cells were incubated in induction media for 24 hr, after which they were harvested and fixed with 4% paraformaldehyde. Following fixation, they were permeabilized with ice-cold 100% methanol and rehydrated with 1% BSA. Cells were then stained for Oct4, Otx2 and Sox2 and analyzed using flow cytometry. For *Snai1* overexpression, cells were seeded at 100,000 cells per 35 mm diameter plate in 2 mL of 3 μM CHIR99021 for 2.5 days. 200 μL of FBS was then added to each plate and 1.8 μg of plasmid was transfected using 5.4 uL of JetPrime (Polyplus). Cells were incubated in transfection media for 12 hr, then washed with PBS and replenished with fresh N2B27 basal media. We then added 3 μL of Tet-Express mixed with 2.5 μL of Intensifier reagent (Clontech). Cells were incubated in induction media for 24 hr, after which they were harvested and fixed with 4%

paraformaldehyde. Following fixation, they were permeabilized with ice-cold 100% methanol and rehydrated with 1% BSA. Cells were then stained for Oct4 and T and analyzed using flow cytometry.

## Fluorescence-activated cell sorting

Cells were trypsinized and fixed in suspension with formaldehyde (4% final concentration, diluted in PBS), permeabilized with ice cold 100% methanol and blocked with 5% donkey serum for 1 hr. Finally, cells are stained with primary antibodies diluted in PBS containing 1% BSA, and detected using fluorescent-tagged secondary antibodies. Flow cytometry was performed on a BD FACSAria flow cytometer equipped with 355 nm, 405 nm, 488 nm, 561 nm, and 637 nm lasers. The data acquired were analyzed using custom programs written in MatLab.

## Generation of mOTX2-Citrine reporter cell line

G4 mESCs, a 129S6 x B6 F1 hybrid line (Andras Nagy, University of Toronto) were maintained on DR4 mouse embryonic fibroblasts (MEFs). These cells ($1 \times 10^7$) were electroporated (Transfection Buffer, Millipore; Bio-Rad set at 250 V and 500 mF) with 5 µg each TALEN plasmid (AI-CN301 and AI-CN302 targeting TTCCAGGTTTTGTGAAGA and TTTAAAAATCACCCACAA, respectively) and 20 µg donor plasmid (AI-CN563). Following transfection, cells were placed on ice for 5 min, then plated onto $3 \times 10$ cm dishes with MEFs. Beginning 30 hr after transfection, cells were selected with hygromycin at 150 µg/mL for 3 days, then 100 µg/mL for an additional 4 days. Approximately 48 hygromycin-resistant colonies were picked and expanded for freezing and DNA preparation and analysis. Five clones were identified with targeted integration by junction PCR (5' junction primers: aagagctaagtgccgccaacagc, catcagcccgtagccgaaggtag; 3' junction primers: cacgctgaacttgtggccgttta, cagctcacctccagcccaaggta). Following expansion and fluorescence-activated cell sorting (FACS), Cerulean$^+$ cells from two clones (2.1 and 2.4) were treated with Cre mRNA. After recovery and expansion, the Cerulean$^-$ cells were enriched by FACS and single-cell cloned. The resulting subclones were tested for removal of the selection cassette (primers: ggtgcctattctggtcgaactggatg, atcacctctgctttgaaggccatgac). The TALENs were kindly provided by the Joung lab synthesized using the FLASH method (*Reyon et al., 2012*). Computation and Modeling were performed using a cluster at Harvard University.

## Software

Calculations were performed using custom written MATLAB code (The Mathworks) on the Harvard Research Computing Odyssey cluster. Code is available at https://github.com/furchtgott/sibilant and https://github.com/sandeepc123/Gene_Regulatory_Network_Modeling respectively. Seurat was done using the package provided in https://github.com/satijalab/seurat (*Macosko et al., 2015*).

# Acknowledgements

We thank Alex Schier, Christof Koch, Ajamete Kayakas, Joshua Levi, Carol Thomson, John Phillips, Paola Arlotta, John Calarco, Leonid Mirny and Andrew Murray for their critical feedback. SJ was funded by the Samsung Scholarship Program. We thank the Allen Institute founders, PG Allen and J Allen and the NIH Directors Pioneer Award 5DP1MH099906-03 and National Science Foundation grant PHY-0952766 for support.

# Additional information

## Funding

| Funder | Author |
| --- | --- |
| Samsung | Sumin Jang |
| NIH Office of the Director | Sharad Ramanathan |
| Office of the Director | Sharad Ramanathan |
| Allen Foundation | Sharad Ramanathan |

The funders had no role in study design, data collection and interpretation, or the decision to submit the work for publication.

## Author contributions

SJ, SC, Conceptualization, Data curation, Formal analysis, Investigation, Methodology, Writing—original draft, Writing—review and editing; LF, Data curation, Formal analysis, Investigation, Methodology, Writing—original draft, Writing—review and editing; L-NZ, Data curation, Methodology; AD, Validation, Methodology; VM, Data curation, Formal analysis; EBL, Data curation, Validation; A-RK, RAM, LM, Resources, Data curation; BPL, Resources, Data curation, Writing—original draft; SR, Conceptualization, Supervision, Funding acquisition, Investigation, Methodology, Writing—original draft, Project administration, Writing—review and editing

## Author ORCIDs

Sumin Jang, http://orcid.org/0000-0001-8918-7059
Leon Furchtgott, http://orcid.org/0000-0002-4258-0950

## Additional files

### Major datasets

The following dataset was generated:

| Author(s) | Year | Dataset title | Dataset URL | Database, license, and accessibility information |
|---|---|---|---|---|
| Sumin Jang, Vilas Menon, Anne-Rachel Krostag, Boaz P Levi, Sharad Ramanathan | 2017 | Dynamics of embryonic stem cell differentiation inferred from single-cell transcriptomics show a series of transitions through discrete cell states | https://www.ncbi.nlm.nih.gov/geo/query/acc.cgi?acc=GSE105054 | Publicly available at the NCBI Gene Expression Omnibus (accession no. GSE105054) |

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
