## [Decision Letter]

Thank you for submitting your article "Dynamics of differentiation inferred from single-cell RNA-seq show a series of transitions through discrete cell states" for consideration by *eLife*. Your article has been favorably evaluated by Arup Chakraborty (Senior Editor) and three reviewers, one of whom, Nir Yosef (Reviewer #1) served as Guest editor. Jacob H. Hanna (Reviewer #3) agreed to share his identity.

The reviewers have discussed the reviews with one another and the Reviewing Editor has drafted this decision to help you prepare a revised submission.

Summary:

In their manuscript, Jang et al. propose, test, and validate a statistical framework for analyzing single-cell transcriptomics data from mouse embryonic stem (mES) cell differentiation. The first part of the analysis relies on a companion manuscript, which presented a combined method for clustering and lineage inference of single cells. By applying this method on data from multiple mouse ES subject to short inductive differentiation protocols, the authors identify several cell states, the genes that mark these states, and the genes that capture state transitions. Within these clusters of cells, the authors assert there is little variation, thus defining discrete cell states along mES cell differentiation. They then cluster the genes into modules and use the Hopfield model to identify patterns of dependencies between modules that give rise to the observed clusters as steady states. With this analysis they provide and validate three hypotheses about possible "rewiring" at different stages (i.e. when the effect of perturbing gene X on gene Y varies between cell states).

Essential revisions:

Overall, the reviewers find the methodology developed in this paper interesting, and of potential impact. However, there are several key points that need to be addressed in order to fully support the validity of this methodology and to understand its intricacies.

1) Single cell data quality:

1.1) Based on the data in Figure 1—figure supplement 1, there seems to be a fairly large variation in the percentage of reads aligning to the transcriptome on a cluster to cluster basis. Do any of the lineages correlate with the percentage of transcriptome or genome reads? What is the significance of the differences between the clusters, like C0 and C3 for example?

1.2) More generally, we are missing a description of how was the RNA-seq data normalized. In many cases when scRNA-seq data is not normalized, we see technical factors that confound the data, and library quality can dominate clustering and dimensionality reduction. Please provide evidence that this is not the case or correct accordingly.

2) Clustering and lineage detection algorithm:

2.1) For their clustering analysis, the authors limit their focus to transcription factors. While they provide off-hand reasoning for this, it is insufficient. Transcription factors (TFs), just like any other transcript, undergo stochastic, burst-like kinetics, and are subject to high amount of variation (esp. given their typically moderate expression). Additionally, it is not a given that measuring TF mRNA, rather than a TF's downstream targets, accurately depicts the circuitry involved in cellular response or differentiation. The authors should demonstrate the effects of including genes other than transcription factors on the clustering results. Relatedly, they later include signaling molecules without a rationale for the shift.

2.2) The nature of the clustering method in which only three clusters of cells are considered at a time inherently limits the hierarchy produced by the author's Bayesian framework (see Figure 2—figure supplement 1, right). In this way, the final lineage tree is limited only to branching into two arms at any given differentiation step. Thus, any differentiation program that produces more than two offspring would not be properly modeled. The authors should address this limitation in their framework.

3) Application to ESC:

3.1) The parameters used for the Bayesian framework from the co-submission are missing. What is the cutoff for a triplet to count as a "transition" event? what is a cutoff for a gene to be defined as a "marker" or "transition" gene? What is the termination/ convergence condition?

3.2) Since the algorithm is iterative, it might be very sensitive to slight variations in initial conditions or the parameters. In standard EM applications, a common practice is to start from many starting conditions. The authors should provide an estimate of how sensitive are the results for the algorithm's parameters (e.g., probability cutoffs) and how sensitive they are for sub-sampling of cells or genes (i.e., going beyond changing the seed set of clusters, which the authors have already done).

3.3) The results in Figure 2, and especially the comparison of 2B vs. 2D are somewhat tautological. It is not clear to me what these figure panels are supposed to show that we don't already know form the definition of the process applied for choosing those genes.

3.4) What is the relationship between the experimental conditions (time/ stimulation; [Supplementary-material SD1-data]) and the inferred clusters? This point is potentially crucial for interpreting the meaning of the clusters and should be discussed.

3.5) We are missing a direct and less engineered view that will help evaluate and digest the clustering results. Specifically – please provide a global heat map figure with all gene used for the final clustering (possibly stratified according to their role as transitions or markers in different parts of the tree) vs. all cells (organized by clusters). This will also help support the statement in the first paragraph of the subsection “Differentiation occurs through a series of discrete cell state transitions”.

3.6) The authors claim that gene expression within each cell cluster does not significantly vary. They validate this by comparing the magnitude of the variance explained by the first PC to the that of the first PC from 1000 sets of randomized data (FYI – unclear how 3B shows lack of significance). Why don't the authors compare the percent variance described by the first PC of each cluster to the percent variance described by first PC of randomized data?

3.7) Can the authors identify early primordial germ cell sub-population (e.g. BLIMP1+, T+, TFAP2C+ cells)? Is it discrete or is it perhaps "hiding" in one of their progenitor populations (e.g. mesendodermal cells)?

4) Validation of results:

4.1) The selection of genes in Figure 3 (immunostaining) seem somewhat biased to well-studied markers (shown in Figure 3—figure supplement 1). Therefore, these results provide a somewhat weak support for the cell states inferred form the single cell data.

4.2) In the subsection “A probabilistic model that replicates the observed discrete cell states predicts state-dependent interpretation of perturbations” the authors mention that they "categorized the 184 marker and transition genes and signaling gene groups into 23 gene modules". However, in Figure 2—figure supplement 2 it seems that the number of transition/ marker genes should be around 800. Also, it is not clear how were the signaling genes selected (since the analysis up to this point focused on transcription factors). Please clarify these points.

5) Network analysis:

5.1) The use of Hopfield model is a nice idea, however the presentation in Figure 4 is somewhat illegible, and it is hard to evaluate the stability of the model (or parts thereof) across the 10k solutions. Please provide a more convenient way to estimate the inferred magnitude and noise for the models parameters. For instance, a scatter plot of parameters showing mean vs. fano factor across the 10,000 solutions; and for a few selected of parameters, the complete empirical distribution.

5.2) How were the gene modules discretized? The explanation in the subsection “1. Determination of gene modules” is insufficient. Specifically – which cutoffs were used? How was gene drop-out taken into account?

5.3) The derivation of the hypotheses (subsection “A probabilistic model that replicates the observed discrete cell states predicts state-dependent interpretation of perturbations”, seventh paragraph) is not defined rigorously. Please describe clearly – what is "effective interaction strength"? How do we decide when "[X] levels are more stable to [Y] overexpression"? Specifically – which statistical cutoffs were used? What is the false discovery rate? How many other, additional hypotheses with a similar FDR can be derived using the same procedure?

---

## [Author Response]

*Essential revisions:*

*Overall, the reviewers find the methodology developed in this paper interesting, and of potential impact. However, there are several key points that need to be addressed in order to fully support the validity of this methodology and to understand its intricacies.*

We would like to thank the Senior and Reviewing Editors and the peer reviewers for their thoughtful comments and suggestions. The feedback and suggestions have greatly improved our manuscript as well as strengthened our conclusions. We have considered each comment and have amended the manuscript to add 3 new figure supplements (20 new subfigures), as well as edited the text as noted in the detailed responses below.

*1) Single cell data quality:*

*1.1) Based on the data in Figure 1—figure supplement 1, there seems to be a fairly large variation in the percentage of reads aligning to the transcriptome on a cluster to cluster basis. Do any of the lineages correlate with the percentage of transcriptome or genome reads? What is the significance of the differences between the clusters, like C0 and C3 for example?*

As the reviewers pointed out, there is a large amount of variation in the total UMI number across cells. Because of this variation and the potential biases it can create, we subsampled an equal number of 20,000 UMI for all 288 cells, and running all subsequent analyses on this subsampled data set. We added a sub-figure (Figure 1—figure supplement 1) to show that following subsampling of UMI’s, cells do not show correlations with one another based on the total number of UMI they had prior to subsampling.

*1.2) More generally, we are missing a description of how was the RNA-seq data normalized. In many cases when scRNA-seq data is not normalized, we see technical factors that confound the data, and library quality can dominate clustering and dimensionality reduction. Please provide evidence that this is not the case or correct accordingly.*

We have added in a sentence (end of subsection “Acquiring single-cell transcriptomics data during early differentiation”) explicitly stating that we normalize our RNA-seq data by subsampling 20,000 UMI’s per cell, for all 288 cells.

*2) Clustering and lineage detection algorithm:*

*2.1) For their clustering analysis, the authors limit their focus to transcription factors. While they provide off-hand reasoning for this, it is insufficient. Transcription factors (TFs), just like any other transcript, undergo stochastic, burst-like kinetics, and are subject to high amount of variation (esp. given their typically moderate expression). Additionally, it is not a given that measuring TF mRNA, rather than a TF's downstream targets, accurately depicts the circuitry involved in cellular response or differentiation. The authors should demonstrate the effects of including genes other than transcription factors on the clustering results. Relatedly, they later include signaling molecules without a rationale for the shift.*

We have added one supplementary figure in the companion manuscript by Furchtgott et al. to address this comment. We analyzed the robustness of using all genes for lineage determination in comparison to using just transcription factors (TFs) and included the results in the accompanying manuscript (Figure 1—figure supplement 2A of accompanying paper Furchtgott et al., 2016).

These figures show a histogram of the number of genes displaying a “clear minimum pattern” (i.e., one cell type within a triplet has a clear minimum expression distribution relative to those of the other two cell types) – evaluated over all TFs and all genes, respectively – from 150 known developmental topologies in B- and T-cell development (Heng et al., 2008). Triplets in which the root has the most genes showing the pattern are labeled red, and triplets in which one of the leaves has the most genes showing the pattern are in blue.

When the gene expression pattern was learned using just TFs (shown on the left side), none of the triplets with more than 10 genes displaying a “clear minimum pattern” exhibit this pattern where the minimum is in the root (no red in any histogram bar except for the leftmost). When the gene expression pattern was learned using just TFs (shown above on the left side), none of the triplets with more than 10 genes displaying a clear minimum pattern have most genes showing a clear minimum in the root (no red in any histogram bar except for the leftmost). However, when all the genes are used for the same analysis (shown above on the right side), even for triplets with a high number (up to 600) of genes showing a clear minimum pattern, a fraction of them have most of these genes showing a clear minimum in the root. This demonstrates that the clear minimum pattern is not robust when using all genes as opposed to just transcription factors.

Further, we added a new figure to this manuscript: Figure 2—figure supplement 3, showing that including all genes for clustering/ lineage determination leads to errors in the inferred lineage tree (although the clustering configuration remains unchanged relative to when the analysis is restricted to only transcription factors). Although the tree remains the same on the mesendodermal branch, towards the bi-potent ectodermal branch the triplet relationships we obtain are different (as well as less parsimonious given the differentiation durations) from the ones we obtain when we use just the TFs.

Lastly, we now explicitly state our rationale (subsection “A probabilistic model that replicates the observed discrete cell states predicts state- 378 dependent interpretation of perturbations”, third paragraph) for adding signaling genes when modeling the gene regulatory network: that it is because our goal of modeling the gene regulatory network is to make specific predictions as to whether and how cells in different states respond differently to perturbations (i.e., *signals* as well as gene expression changes).

*2.2) The nature of the clustering method in which only three clusters of cells are considered at a time inherently limits the hierarchy produced by the author's Bayesian framework (see Figure 2—figure supplement 1, right). In this way, the final lineage tree is limited only to branching into two arms at any given differentiation step. Thus, any differentiation program that produces more than two offspring would not be properly modeled. The authors should address this limitation in their framework.*

We thank the reviewers for bringing up this point. Our framework does *not* in fact inherently limit the resulting lineage tree to bifurcation fate decisions. We illustrate this point using the example in Figure 6.

Author response image 1.**DOI:**
http://dx.doi.org/10.7554/eLife.20487.022

Given the set of triplets shown in Figure 6 (left), the only possible lineage topology is one where the yellow cell type is intermediate to all four other (purple, green, blue, pink) cell types (right). Also, in Figure 2—figure supplement 3, we show an example of a lineage tree that, based on the set of most likely triplets inferred, contains a trifurcation point. Further, in the accompanying manuscript by Furchtgott et. al, the inferred lineage tree from intestinal single- cell data (Figure 3—figure supplement 2) and single-cell human brain development data (Figure 4) contain differentiation steps where three distinct cell types are produced from the same progenitor cell type.

*3) Application to ESC:*

*3.1) The parameters used for the Bayesian framework from the co-submission are missing. What is the cutoff for a triplet to count as a "transition" event? what is a cutoff for a gene to be defined as a "marker" or "transition" gene? What is the termination/ convergence condition?*

The probability cutoff for a triplet to count as a transition event is 0.6. For a gene to be defined as a marker or transition gene, we use a probability cutoff of 0.5. We have added these cutoffs to the manuscript (subsection “Bayesian statistical approach discovers appropriate coordinate systems to infer cell states and state transitions”, fifth paragraph). Further, since the probability cutoffs are arbitrary, in response to this and the following comment we tested our results over a range of cutoffs (see reviewer comment 3.2).

We iterated the clustering-inference procedure until the dimension of the re-clustering subspace i.e., the number of genes changed by less than 10% of the total transcription factor space. We have modified the text (in the sixth paragraph of the aforementioned subsection) to make these points clear.

*3.2) Since the algorithm is iterative, it might be very sensitive to slight variations in initial conditions or the parameters. In standard EM applications, a common practice is to start from many starting conditions. The authors should provide an estimate of how sensitive are the results for the algorithm's parameters (e.g., probability cutoffs) and how sensitive they are for sub-sampling of cells or genes (i.e., going beyond changing the seed set of clusters, which the authors have already done).*

Following the reviewers’ suggestions, we tested the effects of using different probability cutoff values as well as subsampling cells and genes (subsection “Bayesian statistical approach discovers appropriate coordinate systems to infer cell states and state transitions”, seventh paragraph) (Figure 2—figure supplement 3). We show that:

1) The clustering configuration and lineage tree are unchanged within a probability cutoff value range of 0.5 to 0.96 for defining “high-probability” marker and transition genes, but at a cutoff value of 0.97, both the clustering as well as lineage determination given the original clusters fails (Figure 2—figure supplement 3), which we expect to be an effect of the number of genes used for subsequent clustering iterations becoming smaller as the probability cutoff value increases.

2) Further to demonstrating this last point, we show that while using 416 high-probability 𝑝 ≥ 0.96 transition and marker genes still results in the same clustering configuration (with the exception of a few cells, due to the stochastic nature of k- means clustering which becomes more prominent as the number of genes used for clustering becomes smaller) as well as lineage tree. However, using the 416 genes with the highest coefficient of variation across all cells fails to produce the same results but instead gives rise to a much less parsimonious lineage tree, given what we know about the differentiation conditions and duration of the individual cells (Figure 2—figure supplement 3).

3) Finally, we show that our results are robust to using a subset of the 288 genes: the clustering configuration as well as the lineage relationships of the individual clusters to one another remained unchanged when a) a particular cluster (C_3_) was removed, or b) a random subset of 144 cells were removed (Figure 2—figure supplement 3). Interestingly, when C_3_ is removed, the algorithm connects the grandchildren cell types (C_5_ and C_6_) to the grandmother (C_1_) directly.

*3.3) The results in Figure 2, and especially the comparison of 2B vs. 2D are somewhat tautological. It is not clear to me what these figure panels are supposed to show that we don't already know form the definition of the process applied for choosing those genes.*

Figure 2 were intended to be mere illustrations – rather than proofs of principles – of what marker and transition genes are, respectively, although we now see how the comparison between 2B and 2D could appear tautological. Following this comment as well as the next (3.4), we decided to change Figure 2 to a cell-cell correlation plot of all cells using all 889 genes that were used for the final clustering and lineage determination step.

*3.4) What is the relationship between the experimental conditions (time/ stimulation; [Supplementary-material SD1-data]) and the inferred clusters? This point is potentially crucial for interpreting the meaning of the clusters and should be discussed.*

[Supplementary-material SD3-data] explains how the individual cells from each of these wells described in [Supplementary-material SD1-data] cluster. We also added a subfigure (Figure 3—figure supplement 1) to better illustrate the relationship between clusters and culture conditions, showing that cells from different culture conditions sometimes cluster together, as well as that cells exposed to the same conditions are sometimes assigned to different clusters.

*3.5) We are missing a direct and less engineered view that will help evaluate and digest the clustering results. Specifically – please provide a global heat map figure with all gene used for the final clustering (possibly stratified according to their role as transitions or markers in different parts of the tree) vs. all cells (organized by clusters). This will also help support the statement in the first paragraph of the subsection “Differentiation occurs through a series of discrete cell state transitions”.*

We have added a heat map subfigure (Figure 2—figure supplement 2) showing the expression levels of the 899 genes used for the final iteration of clustering and lineage determination for all 288 cells.

*3.6) The authors claim that gene expression within each cell cluster does not significantly vary. They validate this by comparing the magnitude of the variance explained by the first PC to the that of the first PC from 1000 sets of randomized data (FYI – unclear how 3B shows lack of significance). Why don't the authors compare the percent variance described by the first PC of each cluster to the percent variance described by first PC of randomized data?*

We have added a plot showing the mean and c.v. (coefficient of variation) of percent variance explained by the first principal component (PC) of each cell cluster normalized by that of randomized data (Figure 2—figure supplement 3). We find that this value ranges between 1.0400 (C_5_) and 2.2527 (C_0_), with c.v.’s of ~0.01. In contrast, for all possible merged pairs of clusters, the mean and c.v. of the percent variance explained by the first PC normalized by that of randomized data were 3.2220 and 0.2967, respectively. We have referred to this new supplementary figure (subsection “Bayesian statistical approach discovers appropriate coordinate systems to infer cell states and state transitions”, tenth paragraph) in the manuscript.

*3.7) Can the authors identify early primordial germ cell sub-population (e.g. BLIMP1+, T+, TFAP2C+ cells)? Is it discrete or is it perhaps "hiding" in one of their progenitor populations (e.g. mesendodermal cells)?*

We were able to identify one cell in cluster C_7_ that has an above-background expression level of BLIMP1, T, TFAP2C, and STELLA, and although intriguing, given that it is just one cell unfortunately, it is difficult to conclude anything from this observation.

*4) Validation of results:*

*4.1) The selection of genes in Figure 3 (immunostaining) seem somewhat biased to well-studied markers (shown in Figure 3—figure supplement 1). Therefore, these results provide a somewhat weak support for the cell states inferred form the single cell data.*

We have followed up on this comment by immunostaining day 3 and day 4 mesendodermal cells (as identified by T expression) for Etv5 and FoxA2 (Figure 3—figure supplement 1). Etv5 is known to be present in pluripotent cells (Akagi et al., 2015) as well as play a role in spermatogenesis (Tyagi et al., 2009), but to our knowledge it has so far not been implicated with early mesendodermal differentiation.

*4.2) In the subsection “A probabilistic model that replicates the observed discrete cell states predicts state-dependent interpretation of perturbations” the authors mention that they "categorized the 184 marker and transition genes and signaling gene groups into 23 gene modules". However, in Figure 2—figure supplement 2 it seems that the number of transition/ marker genes should be around 800. Also, it is not clear how were the signaling genes selected (since the analysis up to this point focused on transcription factors). Please clarify these points.*

We have added a few sentences in the main text to address these results:

“From the 889 genes that were categorized as either marker or transition genes for all the high probability triplets, we chose genes involving only the triplets along the lineage tree. […] For instance, we did not consider the genes involving the triplet of C_0_, C_1_ and C_5_ since between C_1_ and C_5_ the C_3_ cluster is skipped between C_1_ and C_5_.”

“Further, because our goal was to test whether different cell states were functionally distinct (i.e., respond differently to the same signals and gene expression changes), we also noted the expression pattern of signaling factor genes belonging to FGF, WNT, LIF and BMP signaling pathways along the discovered lineage tree (shown in Figure 3).”

*5) Network analysis:*

*5.1) The use of Hopfield model is a nice idea, however the presentation in Figure 4 is somewhat illegible, and it is hard to evaluate the stability of the model (or parts thereof) across the 10k solutions. Please provide a more convenient way to estimate the inferred magnitude and noise for the models parameters. For instance, a scatter plot of parameters showing mean vs. fano factor across the 10,000 solutions; and for a few selected of parameters, the complete empirical distribution.*

Following the reviewers’ suggestions, we now provide a heatmap showing the mean and c.v. of the different parameters (Jij s) across the 10,000 solutions (Figure 4—figure supplement 2) to provide a convenient way to estimate the inferred magnitude and noise for the model parameters.

*5.2) How were the gene modules discretized? The explanation in the subsection “1. Determination of gene modules” is insufficient. Specifically – which cutoffs were used? How was gene drop-out taken into account?*

We have added the following paragraph to clarify the discretization procedure:

“We obtained 321 transcription factors from the triplets along the tree and classify them based on their pattern across the triplets. […] This way we acquired a total of 29 modules with unique binary gene expression profiles.”

One key assumption of the model is that we can infer the underlying gene regulatory network from the genes that could be detected from the single-cell transcriptomics data. Gene dropout would result in a reduction in the number of genes within a module or a reduction in the number of gene modules. In order to explore how gene dropout would affect our model predictions, we sub-sampled the number of genes to build the Gene regulatory network by changing the probability cutoff for the transition and marker genes we considered. Although the number of gene modules changed (27 modules for a cut off of 0.7 and 24 for 0.9) we found that the models made the same qualitative predictions (Figure 4—figure supplement 3).

*5.3) The derivation of the hypotheses (subsection “A probabilistic model that replicates the observed discrete cell states predicts state-dependent interpretation of perturbations”, seventh paragraph) is not defined rigorously. Please describe clearly – what is "effective interaction strength"? How do we decide when "[X] levels are more stable to [Y] overexpression"? Specifically – which statistical cutoffs were used? What is the false discovery rate? How many other, additional hypotheses with a similar FDR can be derived using the same procedure?*

We would like to thank the reviewers for pointing this out. We have made several changes to this section in order to make it more comprehensible. We have explained the term “effective interaction strength” in the manuscript (subsection “A probabilistic model that replicates the observed discrete cell states predicts state-dependent interpretation of perturbations”, eighth paragraph).

In order to obtain the statistical cutoffs to determine "[X] levels are more stable to [Y] overexpression", we randomly sampled three sets of 3333 of the 10000 models. For each set we computed the relevant quantities, such as the number of models that show downregulation of *Oct4* in response to *Sox2* overexpression, and correspondingly, for each of these quantities computed the mean and the standard error. We have added a few sentences to clarify this point (Figure 4 captions). We have also added these error bars in Figure 4.

Lastly, although our model can make large number of predictions, the eventual confirmation of these predictions can only be done through experimental validation. It is therefore beyond the scope of this manuscript to compute the false discovery rate since that would require experimentally testing hundreds of predictions.